**JGP** Journal of General Physiology

# Structure-function analysis of the lithium-ion selectivity of the voltage-gated sodium channel

Yuki K. Maeda[1,2] ![ORCID], Kentaro Kojima[2] ![ORCID], Tomoe Y. Nakamura[1] ![ORCID], Toru Nakatsu[2] ![ORCID], and Katsumasa Irie[2] ![ORCID]

**Voltage-gated sodium channels (Navs) selectively conduct Na⁺ to generate action potentials. Na⁺ permeates Navs with significantly higher efficiency than many other cations, but Li⁺ can also permeate Navs to an extent comparable with Na⁺. Li⁺ in the blood is known to enter cells via Navs and to have a beneficial effect on various neuropathies. However, the molecular basis of the high Li⁺ selectivity of Navs was unclear. In this study, using a prokaryotic Nav, we successfully created the first Nav mutant to be more selective for Li⁺ than for Na⁺. Electrophysiological and crystallographic analyses suggested the critical determinants of high Li⁺ selectivity: the strong electrostatic interaction between the ion pathway and hydrated ions, and the smaller number of hydration water exchanges within the ion pathway. Additionally, the extensive interactions around the ion pathway were shown to support monovalent cation selectivity. New drug directions based on the molecular basis for Li⁺ permeation may target various neurological disorders and could clarify the broader biological effects of lithium.**

## Introduction

Voltage-gated sodium channels (Navs) are membrane proteins that open in response to depolarization of the membrane potential (Hille, 2001). Navs conduct Na⁺ ions inward quickly ($\sim10^7$ ions/sec) to initiate and propagate rapid action potentials (Catterall, 2023; Hodgkin and Huxley, 1952). The efficiency of Na⁺ permeation of Navs is significantly higher than that of many other cations (Na⁺ ≈ Li⁺ >> K⁺, Cs⁺, and Ca²⁺), so Navs enable the Na⁺ selective translocation across cell membranes in response to stimuli (Campbell, 1976; Hille, 1972; Pappone, 1980). By this property, Navs are key players in nerve conduction, muscle contraction, secretion, neurotransmission, and many other processes.

The high Na⁺ selectivity of Navs is well known, but Navs also conduct lithium ions (Li⁺) with similar efficiency as with Na⁺ (Hille, 2001). Although its physiological role is not widely understood, lithium preparations have been put to use for the treatment of manic, hypomanic, and depressive episodes within bipolar disorder (Shorter, 2009; Hart, 2024; Cade, 1949; Carvalho et al., 2015; Fountoulakis et al., 2016; Malhi et al., 2015; Nestsiarovich et al., 2022; Verdolini et al., 2021). Further beneficial actions of lithium have also been reported, including neuroprotectivity in Alzheimer's disease, reduction of mitochondrial dysfunction, and modulation of apoptosis autophagy (Almeida et al., 2022; Bortolozzi et al., 2024; Chiu et al., 2013; Guttuso et al., 2019; Haupt et al., 2021; Lazzara and Kim, 2015; Leeds et al., 2014; Scheuing et al., 2014; Shim et al., 2023; Lauterbach, 2016; Singulani et al., 2024; Vallée et al., 2021; Vo

et al., 2015). Li⁺ in the blood, liberated from lithium preparations, is assumed to enter cells via Navs (Timmer and Sands, 1999) and is known to affect various cellular systems (Hart, 2024; Schrauzer, 2002). The beneficial effects of lithium are comparatively well understood, but concerns have also been raised about bioaccumulation toxicity, particularly due to increased environmental lithium from industrial products, such as Li⁺ batteries (Chevalier et al., 2024). In vitro, some harmful effects of lithium have been reported, such as neurological, immune, and cardiovascular disorders, as well as fetal diseases (Chevalier et al., 2024). However, the mechanism by which Li⁺ enters the cell, i.e., the molecular basis by which Li⁺ permeates Navs, has not been sufficiently reported.

Na⁺ and Li⁺ are monovalent and single-atom cations, and the ionic radius of Li⁺ (0.60 Å) is smaller than that of Na⁺ (0.95 Å). Due to its small radius, Li⁺ has a higher positive charge density, providing a stronger electrostatic force than Na⁺ (Hille, 2001). Na⁺ and Li⁺ have these different ionic properties, so it is uncertain how they achieve comparable ion permeability in Navs. In mammalian Navs, the pore-forming subunit ($\alpha_1$) consists of four repeats, which are formed by sixfold transmembrane segments (S1–S6) in each repeat (Abriel and Kass, 2005; Catterall, 2000) (Fig. 1 A). Each repeat consists of two domains: S1–S4 detects membrane voltage as the voltage-sensing domain, and S5–S6 forms the ion pore by the segments facing each other, thereby building a pore domain. The ion pore consists of an outer funnel-like vestibule, a selectivity filter (SF), a central cavity,

[1]Department of Pharmacology, Faculty of Medicine, Wakayama Medical University, Wakayama, Japan; [2]Department of Biophysical Chemistry, Faculty of Pharmaceutical Sciences, Wakayama Medical University, Wakayama, Japan.

Correspondence to Katsumasa Irie: kirie@wakayama-med.ac.jp.



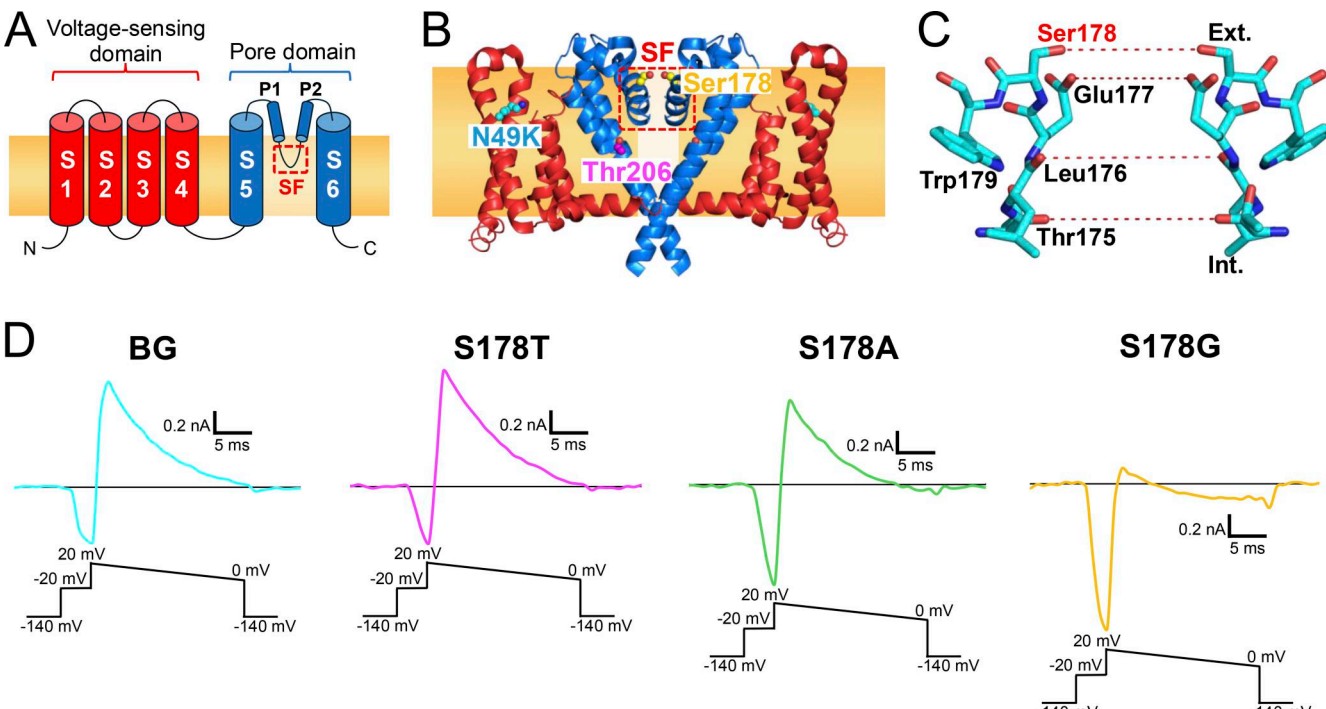

Figure 1. **The SF of NavAb and representative current traces of NavAb N49K, S178T[NK], S178A[NK], and S178G[NK] mutants. (A)** Schematic diagram of the BacNav subunit. Mammalian Navs comprise four homologous repeats. **(B)** Side view of NavAb N49K mutant (PDB code 8H9W). **(C)** Side view of the residues 175–179th constituting the SF of NavAb N49K mutant (PDB code 8H9W). The red dashed lines represent the hydration water-exchange sites through which ions pass, formed by the oxygen atoms of the main and side chains of the SF. **(D)** Recordings of the whole-cell current responses were obtained using the ramp protocol with the solutions of 150 mM $[Li^+]_{out}$ and 150 mM $[Na^+]_{in}$. Background (BG) indicates NavAb N49K mutant without the Ser178 mutation. Currents were generated by the step pulse of –20 mV from the –140 mV holding potential, followed by the ramp pulse with different voltage values (shown at the bottom). In this figure and subsequent electrophysiological measurements shown in Figs. 2, 3, and 9, cancellation of the capacitance transients and leak subtraction were performed using a programmed P/10 protocol delivered at –140 mV.

and an intracellular activation gate. In particular, the SF is especially protruded into the ion pore by the support of the P1 and P2 helices that flank the SF in the pore loop between S5 and S6. The permeation efficiency of each ion is mainly determined by the five amino acid residues of the SF, which narrow the ion pore the most (Fig. 1, A–C) (Hille, 2001). However, the asymmetric and tortuous configuration of vertebrate Navs makes it challenging to fully understand the extent of ion permeation.

As a substitute for vertebrate Navs, the Na⁺ permeation mechanism of Navs is well explained by the functional analyses and three-dimensional structures of several bacterial Navs (BacNavs) (Ahern et al., 2016; Corry and Thomas, 2012; Payandeh et al., 2011). BacNavs comprise homotetramers of subunits with a composition that resembles a repeat of vertebrate Navs. Based on phylogenetic analysis, the lineage of BacNavs likely diverged from a progenitor of eukaryotic Navs and voltage-gated calcium channels (Cavs), and the structural features are conserved among them (Catterall and Zheng, 2015; Liebeskind et al., 2013; Zakon, 2012). Importantly, BacNavs also conduct Li⁺ with similarly high efficiency as eukaryotic Navs (the permeation ratio of Li⁺ over Na⁺ is 0.6–0.8 in BacNavs and is 0.9–1.1 in eukaryotic Navs) (DeCaen et al., 2014; Finol-Urdaneta et al., 2014; Naylor et al., 2016; Hille, 2001). We therefore hypothesized that if

BacNav mutants with enhanced Li⁺ selectivity were created, it might be possible to identify the critical factors of the permeation of Li⁺ common to eukaryotic Navs.

The chemical relations between the strength of the electrostatic force (the "field strength") and the selectivity of monovalent cations have been well established experimentally as the Eisenman sequence, as demonstrated using ion-selective glass electrodes (Hille, 2001; Eisenman, 1962). The Eisenman sequence includes a Li⁺-selective order, but no ion channels have been found that exhibit Li⁺ selectivity. Here, we successfully created the first Li⁺-selective BacNav mutant. Electrophysiology and crystallography suggested two key factors that contribute to high Li⁺ selectivity: the stronger electrostatic force that attracts hydrated ions to the surface of the SF, and the fewer hydration water exchanges that occur during passage through the ion pathway. Furthermore, we showed the dense network around the SF supporting monovalent cation selectivity. Our study is therefore the first to create a Li⁺-selective channel that matches the most Li⁺-selective order in the Eisenman sequence, and we revealed the structural basis of Li⁺ permeation via Navs. This insight into the mechanisms by which Li⁺ enters cells may lead to improved treatments for various neurological disorders and to a deeper understanding of the biological effects of lithium.

## Materials and methods

### Site-directed mutagenesis and construction of NavAb mutants

DNA constructs were produced as previously reported (Irie et al., 2023). The NavAb-mutated DNAs were subcloned into the modified pBiEX-1 vector (71234-3CN; Novagen) that was modified by replacing the fragment from NcoI site (5′-CCATGG-3′) to SalI site (5′-GTCGAC-3′) in the multicloning site with the sequence 5′-CCATGGGCAGCAGCCATCATCATCATCATCACAGCA GCGGCCTGGTGCCGCGCGGCAGCCATATGCTCGAGCTGGTGC CGCGCGGCAGCGGATCCTAAGTCGAC-3′ (Irie et al., 2018). The NavAb-mutated DNAs were subcloned between the BamHI and SalI sites to add a N-terminal His tag and a thrombin cleavage site. Polymerase chain reaction was used to perform site-directed mutagenesis with PrimeSTAR Max DNA Polymerase (Takara Bio). All clones were confirmed by DNA sequencing.

### Electrophysiological measurement in insect cells

The recordings were performed using SF-9 insect cells (ATCC catalog number CRL-1711), which were grown in Sf-900 II medium (Gibco) supplemented with 1% 100× Antibiotic-Antimycotic (Fujifilm-Wako) at 27°C. Cells were transfected with target channel-cloned pBiEX-1 vectors and enhanced green fluorescent protein (EGFP)-cloned pBiEX-1 vectors using polyethyleneimine (PEI) reagent (Cosmo bio). PEI was solubilized in distilled water, and the pH of the 1 mg/ml PEI solution was adjusted to 7.0 with NaOH. First, the channel-cloned vector (3.3 μg) was mixed with 1.65 μg of the EGFP-cloned vector in 150 μl of the culture medium. Next, 15 μl of a 1 mg/ml PEI solution was added, and the mixture was incubated for 10 min before the transfection mixture was gently dropped onto the cultured cells. After 24–48-h incubation, the cells were used for electrophysiological measurements.

Whole-cell recordings were obtained using patch pipettes with resistances ranging from 1.9 to 5.0 MΩ. For reversal potential measurements to determine the relative permeabilities of Na⁺ and other cations, the internal pipette solution contained 35 mM NaCl, 115 mM NaF, 10 mM EGTA, and 10 mM HEPES-NaOH (pH 7.4). The pipette solution was mixed with the acidic solution (35 mM NaCl, 115 mM NaF, 10 mM EGTA, and 10 mM HEPES) and the basic solution (35 mM NaOH, 115 mM NaF, 10 mM EGTA, and 10 mM HEPES) to adjust the pH to 7.4. For the evaluation of cation selectivity, the Li⁺ extracellular solution was mixed with the acidic solution (150 mM LiCl, 10 mM HEPES, 10 mM glucose, and 2 mM CaCl₂) and the basic solution (150 mM LiOH, 10 mM HEPES, 10 mM glucose, and 2 mM CaCl₂) to adjust pH 7.4. The K⁺, Cs⁺, and Ca²⁺ extracellular solutions were similar, except that all LiCl and LiOH were replaced with osmotically equivalent amounts of these cation salts and basic components. The pipette current was zeroed before forming a seal with each cell to account for the 2.25 mV liquid junction potential. Series resistance was compensated and maintained below 10 MΩ before each current recording. Cells with leak current <800 pA were used for data collection, and the mean leak current ranged from 110 to 190 pA for each construct. Cancellation of the capacitance transients and leak subtraction were performed using a programmed P/10 protocol delivered at −140 mV. The bath solution was changed using the Dynaflow Resolve system. All experiments were conducted at 25 ± 1°C using whole-cell patch-clamp mode with a HEKA EPC-10 amplifier and PatchMaster data acquisition software (v2x73). Data export was performed using Igor Pro 9.05 and NeuroMatic (v3.0s) (Rothman and Silver, 2018). All sample numbers represent the number of individual cells used for each measurement. Outliers were excluded if abnormalities were found in other measurement environments, and they were included if no abnormalities were found. All results are presented as mean ± standard error. The graph data were plotted using Microsoft Excel (v16.97).

### Calculation of ion selectivity by the GHK equation

The intracellular and extracellular solutions were arbitrarily set to determine the ion selectivity of each channel. The reversal potential at each concentration was measured by applying a ramp pulse to the membrane potential. A 10-ms depolarization stimulus was inserted for each measurement to confirm the condition of the cell. The permeation ratio of Ca²⁺ and Na⁺ ($P_{Ca}$/$P_{Na}$ ratio) was calculated by substituting the obtained reversal potential ($E_{rev}$) into the expression derived from the GHK equation (Frazier et al., 2000)

$$P_{Ca} \Big/ P_{Na} = \frac{-\left([Na^+]_{in} - [Na^+]_{out}e^{-E_{rev}F/RT}\right)\left(1 - e^{-2E_{rev}F/RT}\right)}{4\left([Ca^{2+}]_{in} - [Ca^{2+}]_{out}e^{-2E_{rev}F/RT}\right)\left(1 - e^{-E_{rev}F/RT}\right)}$$

Here, $R$, $T$, $F$, and $E_{rev}$ are the gas constant, absolute temperature, Faraday constant, and reversal potential, respectively.

The permeation ratio of monovalent cations ($P_M$/$P_{Na}$ ratio, M represents Li, K, or Cs) was calculated by substituting the obtained reversal potential and $P_{Ca}$/$P_{Na}$ ratio into the expression derived from the GHK equation (Lopin et al., 2012)

$$P_M \Big/ P_{Na} = \left[ \frac{-4\left([Ca^{2+}]_{in} - [Ca^{2+}]_{out}e^{-2E_{rev}F/RT}\right)\left(1 - e^{-E_{rev}F/RT}\right)}{\left([Na^+]_{in} - [Na^+]_{out}e^{-E_{rev}F/RT}\right)\left(1 - e^{-2E_{rev}F/RT}\right)} \right.$$

$$\left. \cdot \left(P_{Ca}/P_{Na}\right) - 1 \right] \left[ \frac{\left([Na^+]_{in} - [Na^+]_{out}e^{-E_{rev}F/RT}\right)}{\left([M^+]_{in} - [M^+]_{out}e^{-E_{rev}F/RT}\right)} \right]$$

### Protein expression and purification

Proteins were expressed in the *Escherichia coli* KRX strain (Promega). Cells were grown at 37°C to an OD₆₀₀ of 0.6, induced with 0.1% rhamnose (Fujifilm-Wako), and grown for 16 h at 20°C. The cells were suspended in TBS buffer (20 mM Tris-HCl, pH 8.0, and 150 mM NaCl) and lysed using LAB1000 (SMT Co., LTD.) at 12,000 psi. Low-speed centrifugation removed cell debris (12,000 × *g*, 30 min, 4°C). Membranes were collected by centrifugation (100,000 × *g*, 1 h, 4°C) and solubilized by homogenization in TBS buffer containing 30 mM n-dodecyl-β-D-maltoside (DDM, Anatrace). After centrifugation (40,000 × *g*, 30 min, 4°C), the supernatant was loaded onto a HIS-Select Cobalt Affinity Gel column (Sigma-Aldrich). The protein bound to the cobalt affinity column was washed with 10 mM imidazole in TBS buffer containing 0.05% lauryl maltose neopentyl glycol (LMNG, Anatrace) instead of DDM. After washing, the protein was eluted with 300 mM imidazole, and the His tag was removed by thrombin digestion (overnight, 4°C).

Eluted protein was purified on a Superdex-200 column (Cytiva) in TBS buffer containing 0.05% LMNG.

## Crystallization and structural determination

Before crystallization, the purified protein was concentrated to ~10mg ml$^{-1}$ and reconstituted into a bicelle solution (Faham et al., 2005), containing a 10% bicelle mixture at 2.8:1 (1,2-dimyristoyl-sn-glycero-3-phosphorylcholine [Anatrace]: 3-[(3-cholamidopropyl) dimethylammonio]-2-hydroxypropanesulfonate [DOJINDO]). The NavAb 10% bicelle was mixed in a 20:1 ratio. Prepared proteins were crystallized by sitting-drop vapor diffusion at 20°C by mixing 300-nl volumes of the protein solution (8–10 mg/ml) and the reservoir solution (9–11% polyethylene glycol monomethyl ether [PEG MME] 2000, 100 mM sodium chloride, 100 mM magnesium nitrate, 25 mM cadmium nitrate, and 100 mM Tris-HCl, pH 8.4) with mosquito LCP (STP Labtech). The crystals were grown for 1–3 wk, and after growing, the crystals were transferred into the reservoir solution without cadmium nitrate, which was replaced with the following cryoprotectant solutions. The cryoprotectant solution contains 11% PEG MME 2000, 100 mM Tris-HCl, pH 8.4, 2.5 M sodium chloride, 100 mM calcium nitrate, and 20% (vol/vol) DMSO.

All data were automatically collected at BL41XU and BL45XU of SPring-8 and merged using the ZOO system (Hirata et al., 2019). Data were processed using the KAMO system (Yamashita et al., 2018) with XDS (version 20220110) (Kabsch, 2010). The datasets of NavAb S178T mutants (S178T$^{NK}$ and S178T$^{NK/TA}$) were obtained from a single crystal, while the datasets of NavAb S178A mutants (S178A$^{NK}$ and S178A$^{NK/TA}$) and S178G$^{NK}$ were obtained from four and three crystals, respectively. Analyses of the data with the STARANISO server (Global Phasing Ltd) revealed severely anisotropic crystals. The datasets were therefore ellipsoidally truncated and rescaled to minimize the inclusion of poor-quality diffraction data.

A molecular replacement method using PHASER (McCoy et al., 2007) provided the initial phase using the structure of the NavAb N49K mutant (PDB code 8H9W) as the initial model. The final model was constructed in COOT (version 0.9.2) (Emsley et al., 2010) and refined in REFMAC5 (Murshudov et al., 1997) and Phenix (version 1.18) (Adams et al., 2010). The CCP4 package (version 7.0.078) (Winn et al., 2011) was used for structural analysis. Data collection and refinement statistics for all crystals are summarized in Table S1. All figures were prepared using the program PyMOL 2.5.4 (Schrödinger, LLC, 2015).

## Statistics

Electrophysiological data were analyzed using Microsoft Excel (v16.97). All results are presented as the mean with standard error (SEM). Dunnett's multiple comparison test was used to determine the statistical significance of the NavAb $^{NK/TA}$ Ser178 mutants and NavAb M181A mutants compared with their background constructs, and $P < 0.05$ was considered to be significant. Student's $t$ test was used to estimate statistical significance between the NavAb M181A mutants and their background constructs, and $P < 0.05$ was considered significant. Dunnett's multiple comparison test was performed via the SciPy package

(v1.15.1) in Python (v3.12.8), and Student's $t$ test was performed in Microsoft Excel.

## Online supplemental material

Table S1 is the data collection and refinement statistics of crystallography.

# Results

## Electrophysiological evaluation of ion selectivity

We used NavAb channel, a BacNav cloned from *Arcobacter butzleri*, which is the first full-length structure of Nav at atomic resolution and a helpful model for understanding structure-function relationships of Navs (Payandeh et al., 2011). The wild-type NavAb channel is activated even at very negative membrane potentials, and it requires a holding potential of –240 mV for recovery (Payandeh et al., 2011; Irie et al., 2018). Such a hyperpolarized holding potential complicates the evaluation of channel properties. The N49K mutation in the voltage-sensor domain has been shown to shift the activation potential, allowing a holding potential of –140 mV to maintain channel function (Gamal El-Din et al., 2013; Irie et al., 2018). The N49K site is distant from the SF and does not affect ion selectivity (Fig. 1 B). To ensure stable currents, we therefore introduced the N49K mutation into all NavAb constructs used in this study.

The SF of NavAb consists of "TLESW," in which Ser178 forms the extracellular entrance of the SF (Fig. 1 C) (Payandeh et al., 2011). When Na$^+$ accesses the SF from an extracellular bulk solution, the hydroxyl group of the side chain of Ser178 coordinates with Na$^+$ as the first interaction site in the SF, along with the side chain of Glu177 and four water molecules (Corry and Thomas, 2012). We therefore focused on Ser178 to introduce point mutations. We first examined the Li$^+$ selectivity of the NavAb Ser178 mutants (named S178T$^{NK}$, S178V$^{NK}$, S178A$^{NK}$, and S178G$^{NK}$) in the NavAb N49K mutant (named N49K). Three mutants (S178T$^{NK}$, S178A$^{NK}$, and S178G$^{NK}$) showed current responses in the insect–cell expression system (Fig. 1 D), and S178V$^{NK}$ showed no current. We therefore thought it was impossible to verify larger side-chain mutants than valine. In the Li$^+$ extracellular solution, the reversal potentials of S178T$^{NK}$, S178A$^{NK}$, and S178G$^{NK}$ mutants differed from that of the N49K (Fig. 1 D), which indicated that Li$^+$ selectivity changed by these mutations.

To correctly evaluate ion selectivity, it is necessary to accurately measure the reversal potential. However, measuring the reversal potentials of these mutants was difficult because their currents immediately inactivated. The T206A mutation is known to slow NavAb inactivation and to provide long-lasting currents (Gamal El-Din et al., 2019). NavAb Thr206 is located within the channel lumen, which is distant from Ser178, so the T206A mutation was not expected to influence the effects of mutations at Ser178 (Fig. 1 B). We therefore introduced the T206A mutation to N49K, S178T$^{NK}$, S178A$^{NK}$, and S178G$^{NK}$ mutants (named N49K/T206A, S178T$^{NK/TA}$, S178A$^{NK/TA}$, and S178G$^{NK/TA}$). These NavAb T206A mutants showed slower inactivation and provided more prolonged currents (Fig. 2 A).

Before evaluating ion selectivity, we measured reversal potentials under isoionic conditions for each construct using

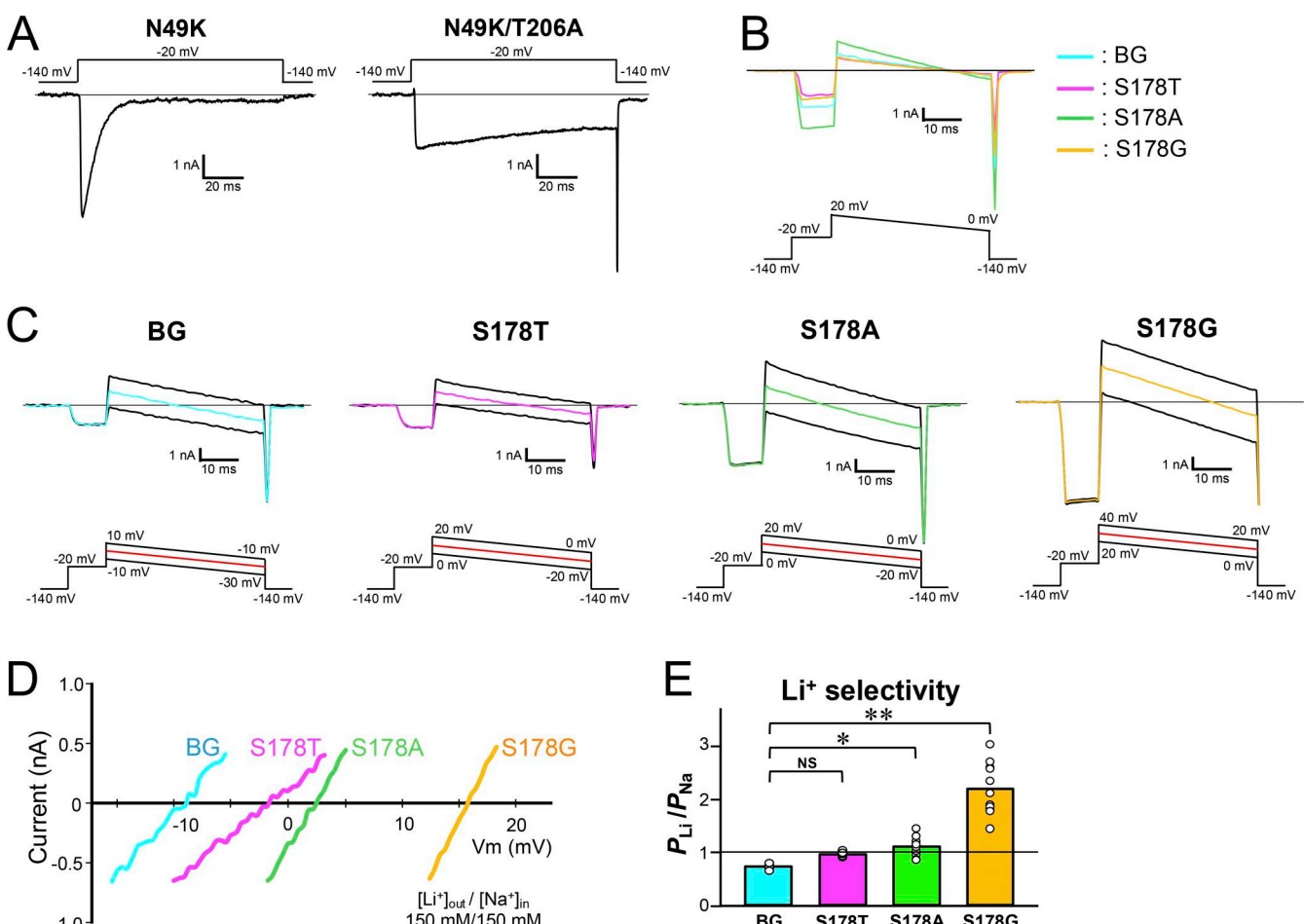

Figure 2. **Li⁺ selectivity evaluation in NavAb N49K/T206A, S178T^{NK/TA}, S178A^{NK/TA}, and S178G^{NK/TA} mutants. (A)** Whole-cell currents in NavAb N49K when a pulse of –20 mV from a –140 mV holding potential was given for 100 ms. **(B)** Current–voltage relationship of each construct with 150 mM $[Na^+]_{out}$ and 150 mM $[Na^+]_{in}$. Background (BG) indicates NavAb N49K/T206A mutant without the Ser178 mutation. Cyan represents BG, and magenta, green, and yellow represent S178T, S178A, and S178G mutants with the BG mutation, respectively. The correspondence between these colors and constructs is the same in C and D. **(C)** Representative current traces of BG and each Ser178 mutant were obtained using the ramp protocol. The reversal potentials recorded with three different ramp pulses were averaged and used. Currents were measured at 150 mM $[Li^+]_{out}$ and 150 mM $[Na^+]_{in}$. **(D)** Current–voltage relationship plots of BG and each Ser178 mutant generated by the ramp pulses in 150 mM $[Li^+]_{out}$ and 150 mM $[Na^+]_{in}$. **(E)** The relative permeability of Li⁺ to Na⁺ in BG and each Ser178 mutant, calculated from the reversal potentials shown in C. Statistical analyses for Li⁺ selectivity among BG and each Ser178 mutant were performed by Dunnett's multiple comparison test. P values <0.05, <0.01, and no significant differences are indicated by "*," "**," and "NS," respectively.

solutions containing 150 mM $[Na^+]_{out}$ and 150 mM $[Na^+]_{in}$ (Fig. 2 B). The reversal potential was +3.98 to +5.20 mV in N49K/T206A, S178T^{NK/TA}, S178A^{NK/TA}, and S178G^{NK/TA}, which was slightly positive due to the presence of 2 mM $Ca^{2+}$ in the external solution. To stabilize the measurement, $Ca^{2+}$ ions were added to the external solution. The positive deviation from 0 mV caused by $Ca^{2+}$ ions remained within one order of magnitude mV, so we considered that 2 mM $Ca^{2+}$ did not disturb the evaluation of the selectivity of other ions.

The permeation ratio of Li⁺ and Na⁺ ($P_{Li}/P_{Na}$ ratio) of N49K/T206A, S178T^{NK/TA}, S178A^{NK/TA}, and S178G^{NK/TA} mutants was evaluated using the reversal potential in a semi-bi-ionic environment. Relatively large currents up to 10 nA were generated in each cell, but series resistance did not affect the membrane voltage at 0 nA, so it was possible to precisely measure the reversal potential. The reversal potential of S178G^{NK/TA} was highest among the other Ser178 mutants, indicating that

S178G^{NK/TA} had the highest Li⁺ selectivity (Fig. 2, C–E). S178A^{NK/TA} also showed a positive shift of reversal potentials relative to N49K/T206A, but to a lesser extent than S178G^{NK/TA}. A slight positive shift was shown by S178T^{NK/TA}, but there was no significant difference in the $P_{Li}/P_{Na}$ ratio compared with N49K/T206A (Fig. 2 E). $P_{Li}/P_{Na}$ ratios were 2.18 ± 0.14 in S178G^{NK/TA} compared with 0.71 ± 0.02 in the N49K/T206A, 0.96 ± 0.01 in S178T^{NK/TA}, and 1.09 ± 0.05 in S178A^{NK/TA} (Table 1). $P_{Li}/P_{Na}$ ratio of S178G^{NK/TA} (2.18 ± 0.14) indicates that the glycine mutant allows Li⁺ to permeate about twice as efficiently as Na⁺, which was approximately triple that of N49K/T206A with Ser178. Next, we also evaluated the selectivity of the other cations (K⁺, Cs⁺, and $Ca^{2+}$). Due to the deactivation induced by the negative membrane potential, it was difficult to perform precise measurements of the reversal potential in these ionic environments. However, the reversal potential around the deactivation membrane potential was sufficient to show that the permeability to

**Table 1. Relative permeability in NavAb N49K/T206A, S178T[NK/TA], S178A[NK/TA], and S178G[NK/TA] mutants**

| | $P_{Li}/P_{Na}$ | $P_K/P_{Na}$ | $P_{Cs}/P_{Na}$ | $P_{Ca}/P_{Na}$ |
|---|---|---|---|---|
| N49K/T206A | $0.71 \pm 0.02$ | <0.05 | <0.05 | <0.05 |
| | ($n = 10$) | ($n = 6$) | ($n = 6$) | ($n = 8$) |
| S178T[NK/TA] | $0.96 \pm 0.01$ | $0.13 \pm 0.01$ | <0.05 | <0.05 |
| | ($n = 8$; P = 0.17) | ($n = 4$) | ($n = 7$) | ($n = 7$) |
| S178A[NK/TA] | $1.09 \pm 0.05$ | <0.05 | <0.05 | $0.12 \pm 0.00$ |
| | ($n = 11$; P = 0.01) | ($n = 8$) | ($n = 6$) | ($n = 6$) |
| S178G[NK/TA] (LivAb[NK/TA]) | $2.18 \pm 0.14$ | <0.05 | <0.05 | <0.05 |
| | ($n = 11$; P = $3.3 \times 10^{-16}$) | ($n = 5$) | ($n = 7$) | ($n = 9$) |

All values are indicated as SEM. When the reversal potential was below the deactivation potential, $P_X/P_{Na}$ ratios (X represents K, Cs, or Ca) were noted as <0.05, which is the value calculated from the lowest measurable reversal potential in each construct. Statistical significances are obtained from Dunnett's multiple comparison test. P values are the statistical significances compared with NavAb N49K/T206A. Statistical tests were not performed for comparisons where $P_X/P_{Na}$ ratios were too low (<0.05).

these ions was extremely low in all constructs. The ratios of $P_K/P_{Na}$, $P_{Cs}/P_{Na}$, and $P_{Ca}/P_{Na}$ remained at low values with little change between N49K/T206A and the other three mutants (Fig. 3, Fig. 4, A and B; and Table 1). In other words, there was no increase in nonselective cation permeability, and only Li$^+$ selectivity was mainly enhanced in these NavAb Ser178 mutants.

**Crystal structure of mutants**
To reveal the molecular mechanisms of the enhancement of Li$^+$ selectivity, S178T[NK], S178A[NK], and S178G[NK] mutants of NavAb were crystallized, and these structures were determined at ∼3.0 Å resolution by the same method as used in our previous study of NavAb (Irie et al., 2018; Irie et al., 2023). Crystallization was also performed in the mutants that contained T206A, but the resolution of these crystals' diffraction patterns deteriorated in all T206A mutants. As with the mutants without T206A, the S178T/T206A mutant showed the best resolution of all T206A mutants, but its resolution was only 3.4 Å (Table S1). The RMSDs of the Cα carbons of the S178T[NK/TA] mutant to S178T[NK] were 0.23 Å. As mentioned earlier, the mutation at Thr206, far from Ser178, is not thought to affect the mutation at Ser178. We therefore performed structural analysis without the T206A mutation. To obtain high-resolution diffraction patterns, crystallization in the presence of Na$^+$ and diffraction experiments in high-Na+ solutions were essential. Unfortunately, we were unable to obtain protein crystals under Li$^+$ conditions without Na$^+$, and when the crystals obtained under Na$^+$ conditions were transferred to a Li$^+$-containing solution, the diffraction patterns disappeared in the diffraction experiments.

The electron density fitted well to the mutated side chain of the 178th residue of all mutants (Fig. 5). Their structures resembled that of the N49K channel (PDB code: 8H9W) (Irie et al., 2023). The RMSDs of the Cα carbons of S178T[NK], S178A[NK], and the S178G[NK] mutants to the N49K were 0.48, 0.52, and 0.46 Å, respectively. The whole structure of each mutant was therefore almost the same as that of the N49K.

The 178th residue is located at the entrance of the ion pathway. Met181 was located on the bulk solution side of the 178th residue. In the NavAb N49K structure, small electron density was observed in this vicinity in the bulk solution (Fig. 6, arrow). In other lithium-selective mutants, larger electron densities were observed in a similar position (Fig. 6, arrowhead and asterisk). These electron densities could come from ions and water molecules contained in the solution. Due to resolution limitations, it is difficult to determine their origin, but it is conceivable that some solution molecules tend to remain in this vicinity. We called these electron densities "vicinal electron densities." In the S178T[NK] and S178A[NK] mutants, the vicinal electron densities were very close to Met181.

Furthermore, in the S178A[NK] mutant, it appears to be one of the rotamers of the Met181 side chain (Fig. 6 C). The vicinal electron densities in the S178G[NK] mutant, the highest Li$^+$-selective mutant, were the largest and closest to the entrance of the ion pathway among the other mutants (Fig. 6 D: asterisk). The glycine mutation created a large space around the 178th residue position due to the absence of side chains (Fig. 6 D). The high Li$^+$ selectivity of the S178G mutant suggested that this wider entrance of the SF contributes to Li$^+$ selectivity. S178A[NK] was the second-highest Li$^+$-selective mutant. The S178G and S178A mutations eliminated the first hydration water-exchange site of the SF formed by the hydroxyl groups of Ser178. The smaller number of hydration water-exchange sites would be a candidate for the Li$^+$ selectivity promoter. At the same time, smaller side-chain mutations exposed the negatively charged side chain of Glu177, and this exposed and negatively charged side chain would also be a candidate for the Li$^+$-selectivity promoter.

**The shape of the ion pathway**
As the size of the mutated side chain increased, the radius of the ion pore decreased (Fig. 7). In the S178T[NK] mutants, the mutated side chain was the narrowest region, even within the SF, with a length <2 Å (Fig. 7, B and E). The wild-type Ser residue forms a similarly narrow region, but the radius of this ion pore is >2 Å. In the S178A[NK] and S178G[NK] mutants, the shape of the ion pore in this region is more expansive because of the smaller side-chain mutation (Fig. 7, C and D). These two mutants exhibit improved

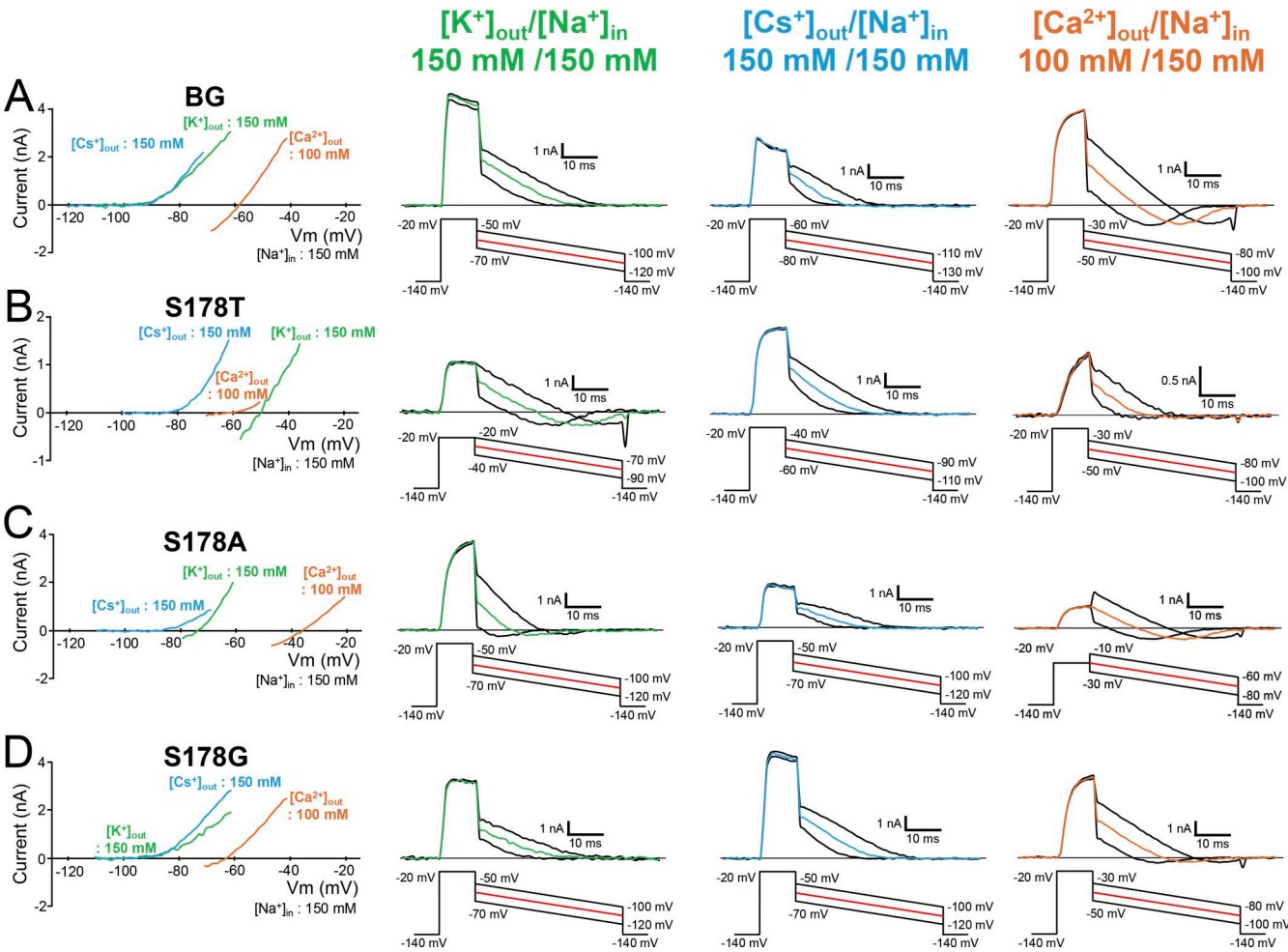

Figure 3. **Cation selectivity evaluation in NavAb N49K/T206A, S178T[NK/TA], S178A[NK/TA], and S178G[NK/TA] mutants. (A–D)** Current–voltage relationship plots (left) and representative currents (right) used to evaluate the permeability of K+, Cs+, and Ca2+ relative to that of Na+. Background (BG) indicates NavAb N49K/T206A mutant without the Ser178 mutation. Currents were measured at 150 mM [K+]out (green), 150 mM [Cs+]out (sky blue), and 100 mM [Ca2+]out (orange), respectively, and 150 mM [Na+]in. Currents were generated by a step pulse from a –140 mV holding potential to –20 mV, followed by ramp pulses to different voltage values. The time course of the change of membrane potentials is shown at the bottom of the respective current traces.

Li+ selectivity, suggesting that the Li+ selectivity has been enhanced by widening the pore. On the other hand, this is inconsistent with S178T[NK] (which has narrower pores) showing a slight improvement in Li+ selectivity, although without a significant difference.

Next, we focused on residues that either seem likely to mediate interactions with ions as they transit the pore or that are thought likely to help stabilize SF structure (Fig. 8). In the NavAb SF, three residues, "176-LES-178," form the surface of the ion pathway (Fig. 7). The Met181 at the P2 helix was located on the extracellular side of the 178th amino acid residue. In the S178G[NK] case, the Met181 side chain moved toward the Gln172 side chain at the P1 helix of the neighboring subunit (Fig. 8 D). Gln172, which is located in the backyard of the ion pore, does not face the ion permeation pathway, but its side chain forms a hydrogen bond with the carbonyl group of Glu177 main chain (Fig. 8). The side chain of Glu177, a carboxylate, forms a critical ion recognition site for ion permeation. The side chain of Arg185 forms a hydrogen bond with the oxygen atom of the side chain of Gln172

and fixes the direction of the amido group of Gln172 to stabilize the hydrogen bond between the Glu177 main chain and the Gln172 amido group. This interaction network involving the three residues was also observed in all mutants (Fig. 8, A–D). Although resolution was limited, the hydrogen bond between the main chain of glutamate 177 and the side chain of glutamine 172 appeared closer in the S178G mutant than in other variants (Fig. 8 D). We focused on glutamine 172, which mediates the hydrogen bond network, as a regulator of ion permeability and selectivity. Gln172 of NavAb N49K/T206A and S178G[NK/TA] were mutated to evaluate their role in regulating Li+ selectivity. The asparagine side chain is shorter than that of glutamine by one methyl group deletion, so the Q172N mutation might be expected to weaken or abolish the hydrogen bond networks with Glu177 and Arg185. In five cells per mutant from a single transfection, we failed to observe any current with the NavAb Q172N mutants. This fatal loss of function due to a slight difference in side-chain length suggested a critical role for Gln172 of NavAb in ion permeability and stability of the SF.

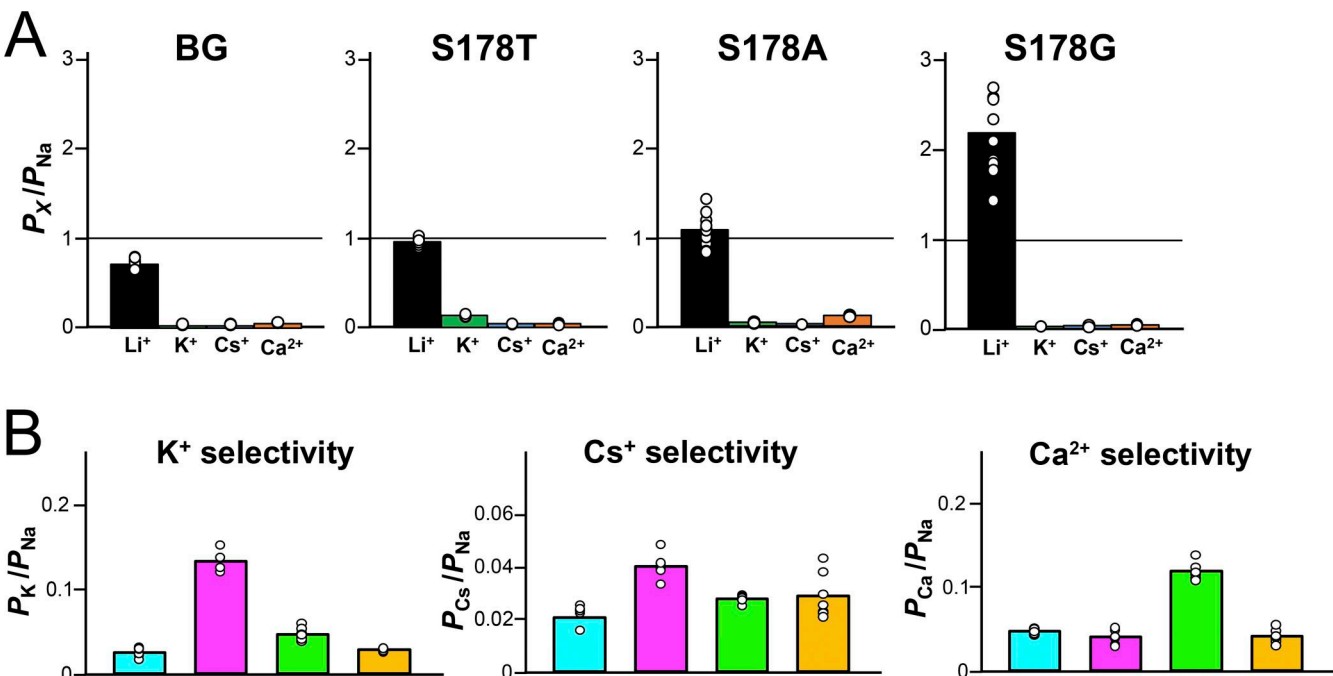

**Figure 4. Cation selectivity evaluation and statistical analyses in NavAb N49K/T206A, S178T[NK/TA], S178A[NK/TA], and S178G[NK/TA] mutants. (A)** The relative permeability of each cation to Na+ ($P_X/P_{Na}$ ratios, X represents Li, K, Cs, or Ca) in NavAb N49K/T206A and each Ser178 mutant, calculated from the reversal potentials that were obtained by the ramp pulses shown in Fig. 2 C and Fig. 3, A–D. Background (BG) indicates N49K/T206A mutant without the Ser178 mutation. **(B)** The selectivity for K+, Cs+, and Ca2+ of BG and each Ser178 mutant.

## Effects of mutation of a pore helix residue on monovalent cation selectivity

An increase in the vicinal electron density near Met181 was a common feature observed in the mutants showing enhanced Li+ selectivity (Fig. 6, B–D). In the S178A[NK] mutant, it even appears to be one of the rotamers of the Met181 side chain (Fig. 6 C). The mechanism is unclear, but an alternative conformation of the Met181 side chain might improve Li+ selectivity. To evaluate the role of Met181 in the ion selectivity, we therefore introduced the M181A mutations to the NavAb N49K/T206A and its Ser178 mutants (named N49K/T206A-M181A, S178T[NK/TA]-M181A, S178A[NK/TA]-M181A, and S178G[NK/TA]-M181A). Cells expressing S178T[NK/TA]-M181A showed little current and sometimes a

current too weak (<1 nA) to evaluate reversal potential (Fig. 9 A). Additionally, cells exhibiting channel current were scarce in S178A[NK/TA]-M181A and S178G[NK/TA]-M181A, but we nonetheless obtained a credible reversal potential. The resulting $P_{Li}/P_{Na}$ ratios were 0.85 ± 0.03 in N49K/T206A-M181A, 1.09 ± 0.04 in S178A[NK/TA]-M181A, and 0.99 ± 0.04 in S178G[NK/TA]-M181A, respectively (Fig. 9 B and Table 2). Notably, the enhanced Li+ selectivity observed in S178G[NK/TA] was attenuated by the M181A mutation, showing no significant difference from the N49K/T206A mutant (Fig. 9 C). The Li+ selectivity of S178A[NK/TA]-M181A mutant was almost unchanged from that of S178A[NK/TA] mutant. The $P_{Li}/P_{Na}$ ratios of those mutants approached 1, indicating nonselectivity for Li+ and Na+. $P_{Ca}/P_{Na}$ ratios significantly

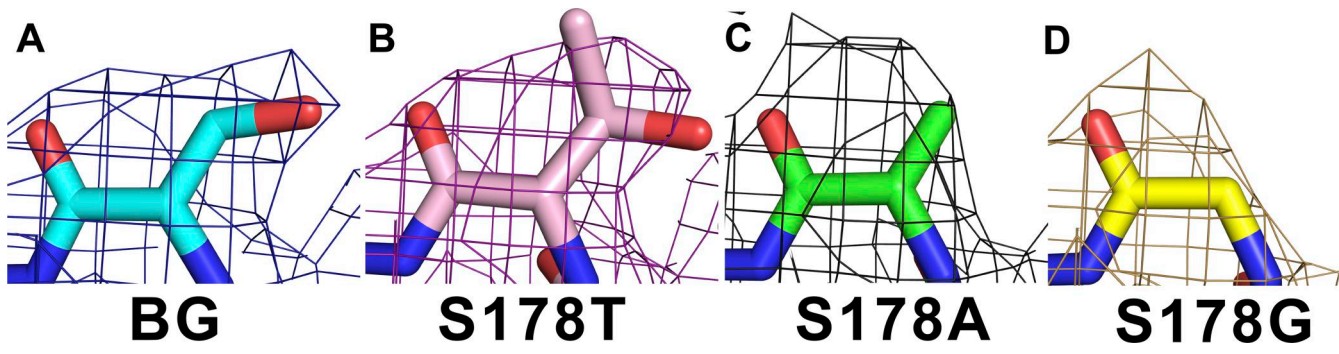

**Figure 5. The difference density map of mutational residues. (A–D)** An *Fo-Fc* electron density map with the omitted amino acid residues surrounding the SF (172nd residue to 185th residue), focused on the 178th residue position of NavAb N49K, S178T[NK], S178A[NK], and S178G[NK] mutants, respectively. Background (BG) indicates N49K mutant without the Ser178 mutation. Each mesh indicates the electron density maps contoured at 3σ.

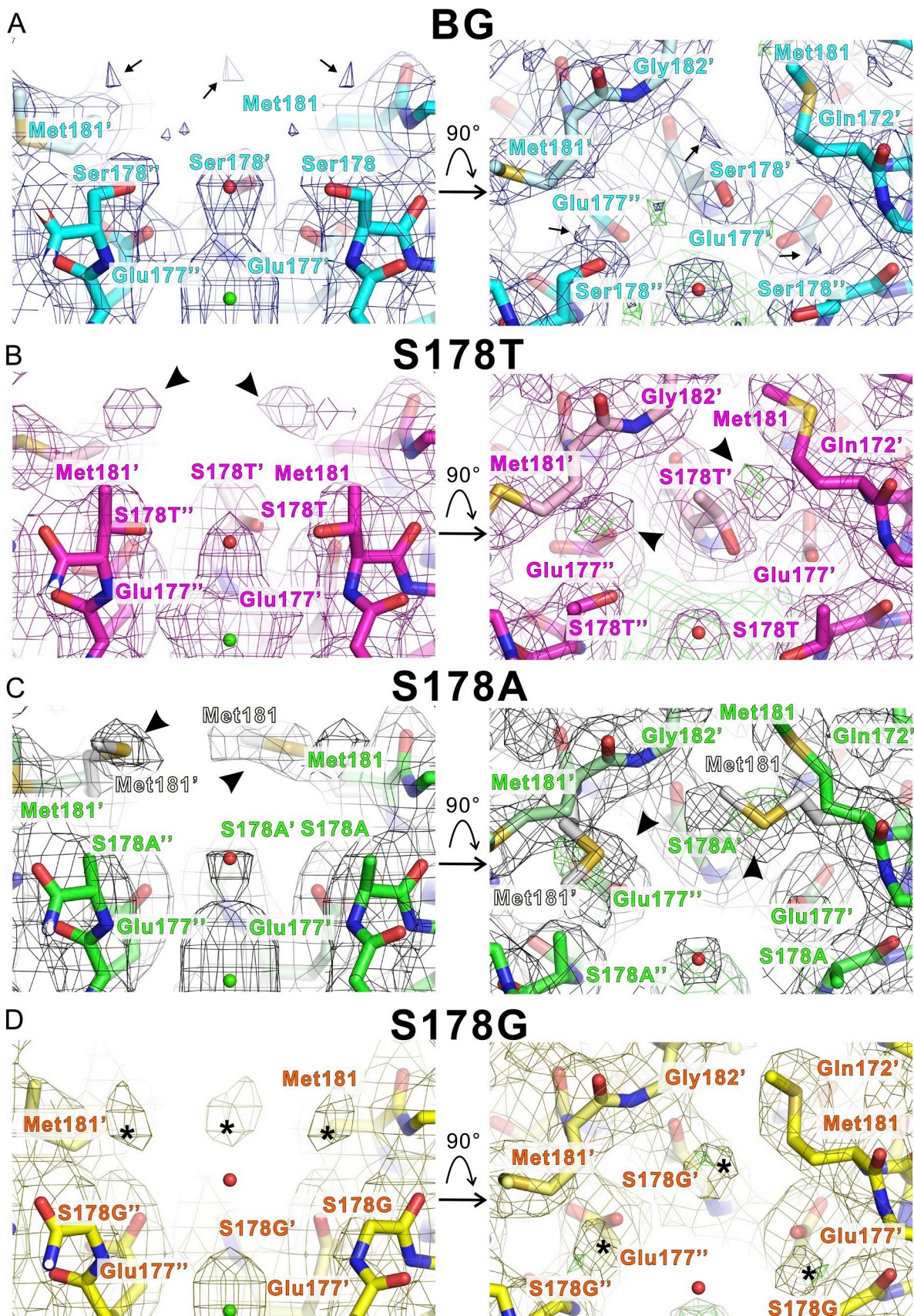

Figure 6. **The SF structure of NavAb N49K, S178T[NK], S178A[NK], and S178G[NK] mutants in high sodium conditions. (A–D)** Horizontal (left side) and vertical (right side) views from the extracellular side of the ion pathway of NavAb N49K, S178T[NK], S178A[NK], and S178G[NK] mutants with electron density map,

respectively. Background (BG) indicates N49K mutant without the Ser178 mutation. The carbon atoms of NavAb N49K, S178T[NK], S178A[NK], and S178G[NK] mutants are colored cyan, magenta, green, and yellow, respectively. The 2Fo-Fc electron density of NavAb N49K, S178T[NK], S178A[NK], and S178G[NK] mutants is colored blue, dark magenta, grey, and orange, respectively. Black arrows indicate the small vicinal electron densities. Black arrowheads indicate the large vicinal electron density connected to the Met181 side chain. An asterisk indicates the large vicinal electron density isolated from the Met181 side chain. The green particles indicated the calcium ions. The residues of the center subunit are numbered with prime symbols. The residues of the left subunit are numbered with double prime symbols. In vertical view, Fo-Fc electron density map is colored green and contoured at 3σ. In C and G, the dual conformation side chain of Met181 was depicted as a white stick.

increased by the M181A mutation compared with the background mutants (Fig. 10). These results indicated that the M181A mutations caused a loss of monovalent cation selectivity and of divalent cation exclusion. Met181 of NavAb was thus implied to support the stabilization of the SF and the selection of monovalent and divalent cations.

## Discussion

Various ion selectivity analyses using BacNavs have been reported, but Li+ selectivity in all mutants was almost unchanged from that of wild types ($P_{Li}/P_{Na}$ ratios = 0.6–0.8 in wild types and their mutants) (DeCaen et al., 2014; Finol-Urdaneta et al., 2014; Naylor et al., 2016). In this study, the Ser178 mutations of NavAb were found to enhance Li+ selectivity. NavAb S178G[NK/TA], with the highest Li+ selectivity exceeding Na+ selectivity (Li+ > Na+ > K+, and Cs+), is thought to be the first channel that exactly belongs to the most Li+-selective order of the Eisenman sequence. Following the nomenclature of

BacNavs (Yue et al., 2002; Tang et al., 2014), the NavAb S178G mutants were henceforth collectively referred to as "LivAb," based on their high Li+ selectivity (S178G[NK] and S178G[NK/TA] were named LivAb[NK] and LivAb[NK/TA], respectively).

### Effects of changes in hydration water exchanges in the SF

Hydration water exchanges are crucial for ion permeation in Navs (Ahern et al., 2016). The diameter of the SF of Navs is sufficiently wide that two to three Na+ ions are accommodated in the SF and permeate in a hydrated or partially dehydrated state (Chakrabarti et al., 2013; McCusker et al., 2012; Payandeh et al., 2011; Shaya et al., 2014; Tsai et al., 2013; Ulmschneider et al., 2013). Referring to the crystal structure of NavAb, the oxygen atoms in the main and side chains in the SF form four hydration water-exchange sites (Fig. 1 C) (Payandeh et al., 2011). When fully hydrated Na+ ions in bulk solution approach the outer edge of the SF in BacNavs, they first interact with the protein via water molecules (Naylor et al., 2016). In the case of NavAb, entering into the SF requires the Na+ to

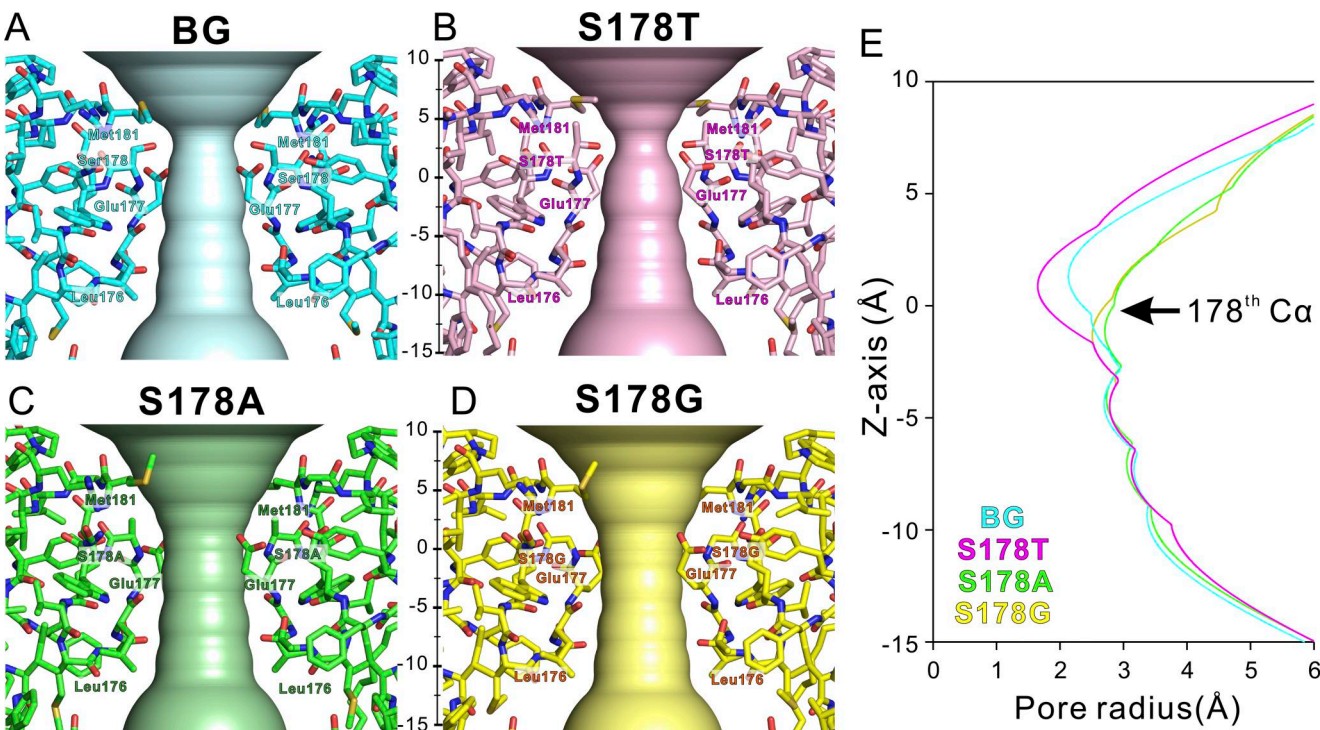

Figure 7. **The shapes and radius of the ion pathway of NavAb N49K, S178T[NK], S178A[NK], and S178G[NK] mutants. (A–D)** Horizontal view of the ion pathway of NavAb N49K, S178T[NK], S178A[NK], and S178G[NK] mutants depicted by program hole. Background (BG) indicates N49K mutant without the Ser178 mutation. The carbon atoms of NavAb N49K, S178T[NK], S178A[NK], and S178G[NK] mutants are colored cyan, magenta, green, and yellow, respectively. **(E)** The pore radius of NavAb N49K, S178T[NK], S178A[NK], and S178G[NK] mutants. The baseline of the z axis is set to the z-coordinate of the main-chain carbonyl oxygen atoms of the 178th residue. The extracellular direction is plotted as plus, and the cytosolic direction as minus.

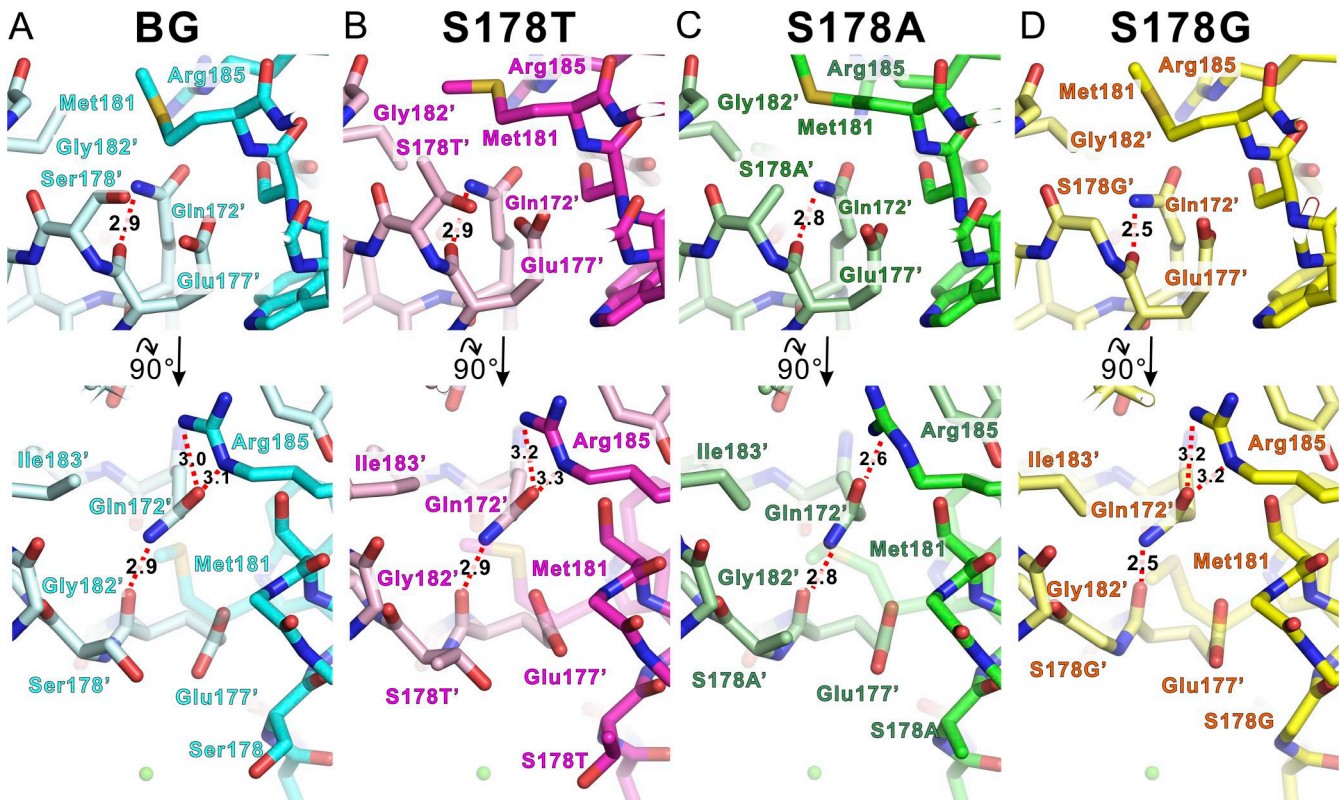

**Figure 8.** **The hydrogen bond network in the backyard of the SF. (A–D)** Upper: The horizontal view from the center of the ion pathway of the protein structures around Met181 and adjacent subunit's Glu177 and Gln172 of the SF of NavAb N49K, S178T[NK], S178A[NK], and S178G[NK] mutants, respectively. Background (BG) indicates N49K mutant without the Ser178 mutation. These figures correspond to the enlarged view of the subunit at the back center in Fig. 6. Lower: The vertical view from the extracellular side of the protein structures around Met181, indicating Arg185 interaction was lost in S178G[NK]. The residues of the left subunit are numbered with a prime symbol.

release two hydration waters, which are subsequently replaced by the oxygen atoms of the side chains of Ser178 and Glu177 (Corry and Thomas, 2012). Na[+] passes through the hydration water-exchange sites formed by the main-chain carbonyls of Leu176 and Thr175 and then enters the central cavity (Payandeh et al., 2011; Chakrabarti et al., 2013). Changes in the number and location of hydration water exchanges in the SF will greatly affect the permeation efficiency of ions with different properties.

Hydration water exchange is one of the rate-limiting steps of ion transfer. Li[+] (0.60 Å) has a smaller ionic radius than Na[+] (0.95 Å), resulting in a higher positive charge density and a stronger electrostatic interaction with the negative dipole of water molecules and proteins. Consequently, the rate of hydration water exchange for Li[+] is slower than that for Na[+] (Hille, 2001). This means that with a greater number of hydration water-exchange sites formed by hydroxyl groups in the SF, there will be a slower rate of Li[+] permeation due to strong interaction forces and lower permeability. In this study, the hydroxyl groups that form the first hydration water-exchange site in the SF were lost in NavAb S178A[NK/TA] and S178G[NK/TA] (LivAb[NK/TA]) compared with N49K/T206A (Fig. 11). This decrease in hydration water exchanges in the SF may have favored the permeation of Li[+], which has a slower exchange rate.

**Enhanced electrostatic interactions are the most promising candidate for increasing Li[+] selectivity**

The reduction of hydration water exchanges in the SF alone would not be expected to increase Li[+] permeability exceeding that of Na[+] because the permeability of both Na[+] and Li[+] is both increased simultaneously. According to the Eisenman sequence, which divides the monovalent cation selectivity into 11 sequences, the selectivity of Li[+] only exceeds that of Na[+] when the electrostatic force interacting with ions is most potent (Eisenman sequence XI: Li[+] > Na[+] > K[+] > Cs[+]) (Hille, 2001; Eisenman, 1962). The most potent electrostatic interaction site of NavAb is comprised of the side chain of Glu177, which is only negatively charged in the SF (Fig. 1 C) (Payandeh et al., 2011). Glu177 of NavAb and corresponding residues of eukaryotic Navs and Cavs are known to bind cations as the most strongly electrostatic site and facilitate selective permeation of ions (Boiteux et al., 2014; Chakrabarti et al., 2013; Favre et al., 1996; Ke et al., 2014; Lipkind and Fozzard, 2000; Xia et al., 2013). In NavAb S178A[NK/TA] and S178G[NK/TA] (LivAb[NK/TA]) mutants, which had greatly improved Li[+] selectivity compared with N49K/T206A (Fig. 2 E and Table 1), the loss of the hydration water exchange site formed by the side chain of Ser178 would expose Glu177 directly to the bulk solution (Fig. 6, C and D; and Fig. 11). The strength of an ionic bond depends on the electrostatic force and, inversely, on the sum of the radii of two particles. The exposed electrostatic site is therefore

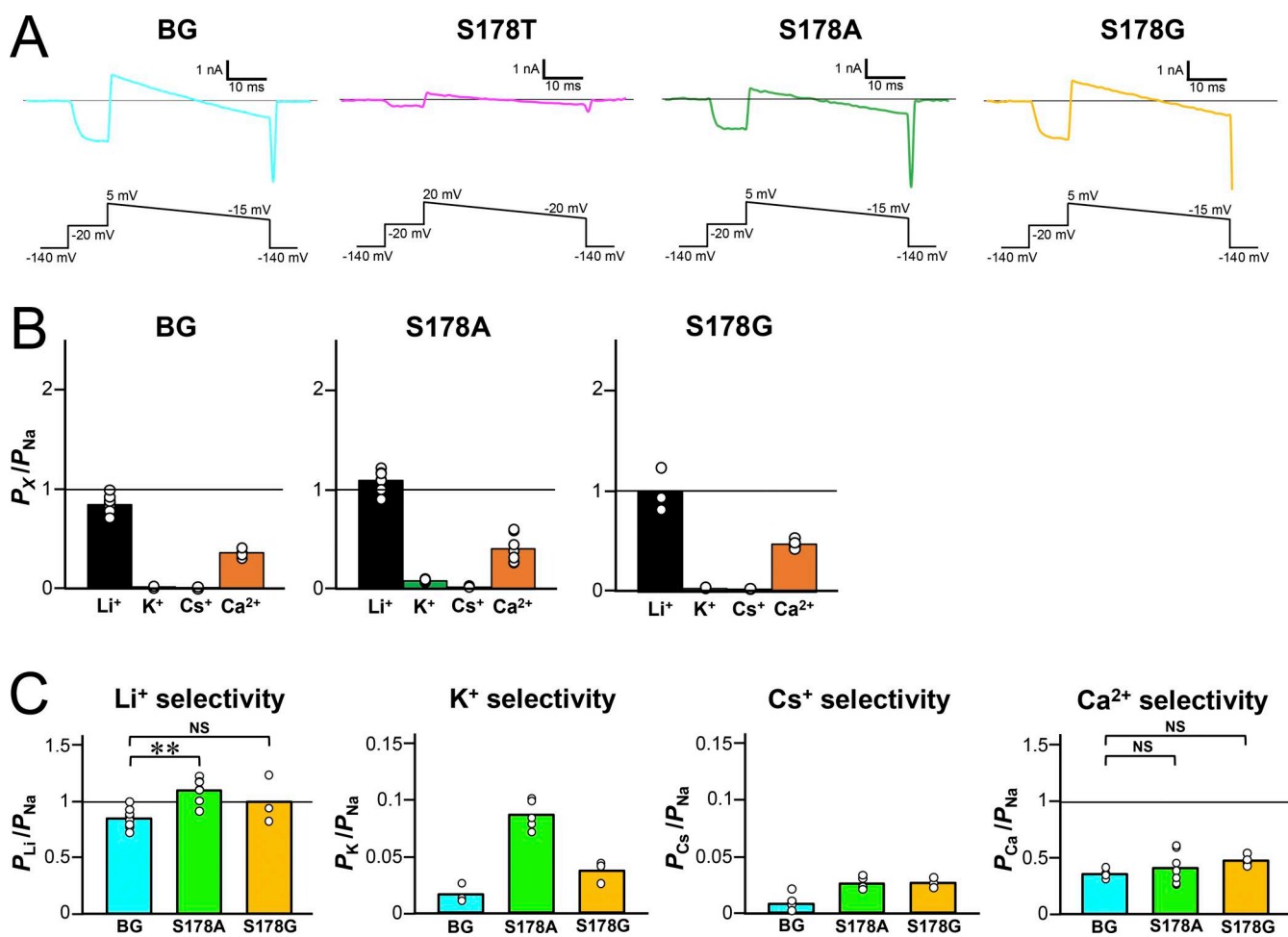

Figure 9. **Cation selectivity evaluation in NavAb N49K/T206A-M181A, S178A^NK/TA^-M181A, and S178G^NK/TA^-M181A mutants. (A)** Representative current traces obtained from the ramp protocol in the solutions of 150 mM $[Li^+]_{out}$ and 150 mM $[Na^+]_{in}$. Background (BG) indicates N49K/T206A/Met181 mutant without the Ser178 mutation. **(B)** The relative permeability of each cation to $Na^+$ ($P_X/P_{Na}$ ratios, X represents Li, K, Cs, or Ca) in each NavAb M181A mutant, calculated from the reversal potentials that were obtained by the ramp pulses shown in A. **(C)** Cation selectivity of BG and each Ser178 mutant, with the results of Dunnett's multiple comparison test. P values <0.01 and no significant differences are indicated by ** and NS, respectively.

advantageous for ionic bonding with $Li^+$, which has a smaller ionic radius and a higher positive charge density than $Na^+$. The exposed Glu177 side chain is related to enhanced $Li^+$ selectivity, which corroborates the correctness of the Eisenman sequence.

In addition, in the NavAb S178G (LivAb), with the glycine mutation widening the SF vestibule (Fig. 6 D and Fig. 11), ions from bulk solution would be able to easily approach the exposed electrostatic site directly, leading to a reversal of $Li^+$ and $Na^+$

Table 2. **Relative permeability in NavAb Met181 mutants**

| | $P_{Li}/P_{Na}$ | $P_K/P_{Na}$ | $P_{Cs}/P_{Na}$ | $P_{Ca}/P_{Na}$ |
|---|---|---|---|---|
| N49K/T206A-M181A | 0.84 ± 0.03 | <0.05 | <0.05 | 0.36 ± 0.02 |
| | ($n$ = 8) | ($n$ = 3) | ($n$ = 6) | ($n$ = 4) |
| S178A^NK/TA^-M181A | 1.09 ± 0.04 | 0.09 ± 0.01 | <0.05 | 0.41 ± 0.05 |
| | ($n$ = 7; P = 2.2 × 10⁻³) | ($n$ = 5) | ($n$ = 5) | ($n$ = 7; P = 0.63) |
| S178G^NK/TA^-M181A (LivAb^NK/TA^-M181A) | 0.99 ± 0.04 | <0.05 | <0.05 | 0.47 ± 0.05 |
| | ($n$ = 3; P = 0.16) | ($n$ = 3) | ($n$ = 3) | ($n$ = 4; P = 0.24) |

All values are indicated as SEM. When the reversal potential was below the deactivation potential, $P_X/P_{Na}$ ratios (X represents K, Cs, or Ca) were noted as <0.05, which is the value calculated from the lowest measurable reversal potential in each construct. Statistical significance obtained from Dunnett's multiple comparison test. P values are the statistical significances compared with NavAb N49K/T206A-M181A. Statistical tests were not performed for comparisons where $P_X/P_{Na}$ ratios were too low (<0.05).

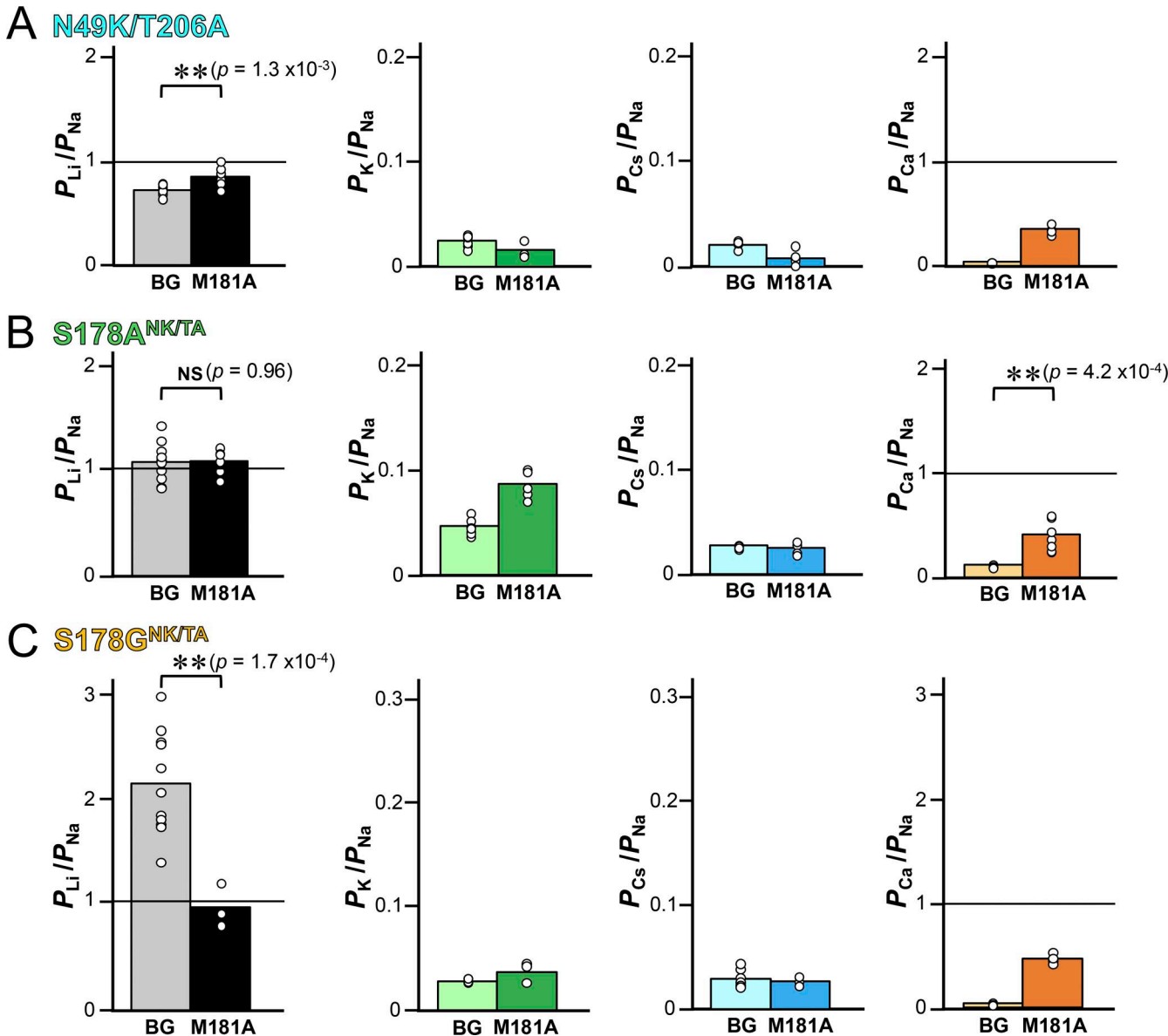

Figure 10. **The results of Student's *t* test for cation selectivity among NavAb M181A mutants and their mutational backgrounds. (A–C)** Background (BG) indicates constructs without the M181A mutation (N49K/T206A, S178A[NK/TA], and S178G[NK/TA]). Statistical analyses were performed by Student's *t* test. P values <0.01 and no significant differences are indicated by ** and NS, respectively.

permeability. In the crystal structures of S178A[NK] and S178G[NK] (LivAb[NK]), the vicinal electron density was observed at the SF vestibule (Fig. 6, C and D). If the vicinal electron densities were water molecules, it would explain the high accessibility of hydrated ions in bulk solution to the most electrostatic sites in these mutants.

**P-helices stabilize the SF and support ion selectivity in NavAb**
Here, we examine why the NavAb S178G[NK/TA] mutant retains low Ca²⁺ selectivity, with $P_{Ca}/P_{Na}$ ratio <0.01, while enhancing Li⁺ selectivity, despite Ca²⁺ having a stronger electrostatic force than monovalent cations. We previously reported on the only identified prokaryotic Cav, CavMr (Irie, 2021; Shimomura et al., 2020). CavMr functions as a

homotetramer, similar to prokaryotic Navs, and the SF sequence of CavMr is "TLEGW," the same as that of S178G[NK/TA] (LivAb[NK/TA]). The glycine residue in the SF (TLEGW) of CavMr plays a key role in Ca²⁺ selectivity, and a single-point mutation of its glycine to serine decreases $P_{Ca}/P_{Na}$ ratio by about 18-fold. In contrast, in the current study, we found that the S178G mutation of NavAb affected only Li⁺ selectivity, not Ca²⁺ selectivity (Fig. 4, A and B; and Table 1). However, all the NavAb M181A mutants were converted into nonselective channels with moderate Li⁺ permeability, equivalent to Na⁺, and ~10-fold Ca²⁺ selectivity (Fig. 9, B and C; and Table 2). The Met181 of NavAb is suggested by this result to be essential in sorting monovalent and divalent cations (Li⁺ and Na⁺ vs. Ca²⁺ in this case).

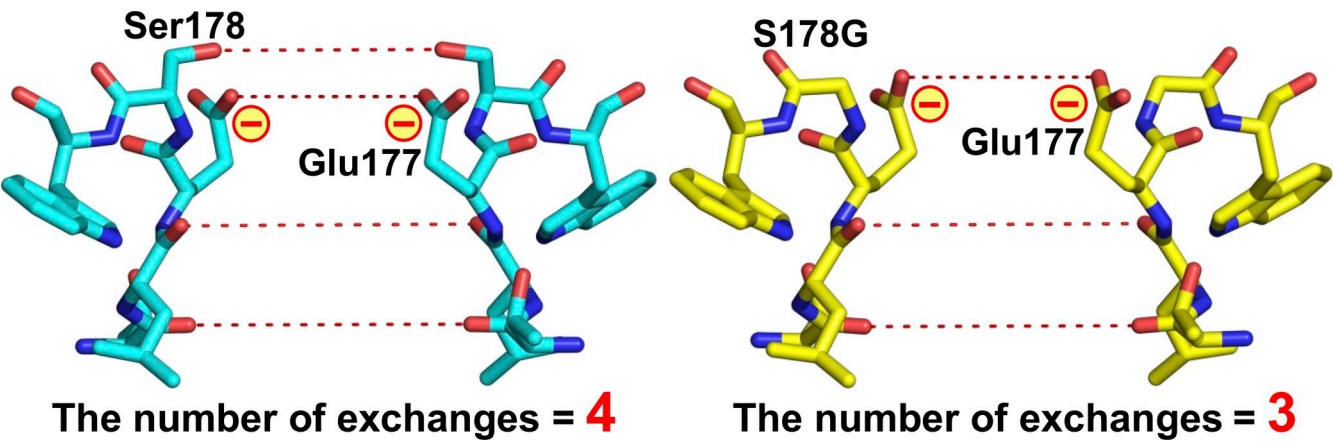

**Figure 11.** **The SF of NavAb N49K and S178G^NK (LivAb^NK) mutants.** Side view of the residues 175–179th constituting the SF of N49K (left, PDB code 8H9W) and S178G^NK (LivAb^NK) (right, PDB code 9UC3). The red dashed lines represent the hydration water-exchange sites through which ions pass, formed by the oxygen atoms of the main and side chains of the SF.

The Met181 of NavAb is Ser in NaChBac, which is one of the representative BacNavs (Ren et al., 2001; Yue et al., 2002). Ser178 is highly conserved in BacNavs, but a similar glycine mutation of other BacNavs, such as S178G of NavAb, may not improve Li⁺ selectivity. The corresponding position of NavAb Met181 in CavMr, which is the non-artificial bacterial Cav, is aspartate (Shimomura et al., 2020), and the same is true for artificial Cavs (CaChBac and CavAb) (Yue et al., 2002; Tang et al., 2014). Their aspartate side chains are thought to affect cation permeability through electrostatic interactions. On the other hand, if only electrostatic repulsion is required for cation selectivity by Met181 in NavAb, Li⁺ should also be excluded due to its higher positive charge density compared with Na⁺, as should Ca²⁺. In other words, factors other than electrostatic interaction are involved in the monovalent cation selectivity by NavAb Met181. Around the SF, extensive interactions are formed within the same subunit or between neighboring subunits (Payandeh et al., 2011). As for the crystal structure of NavAb, it is known that Met181 and Gln172, located on the pore helices, also form part of the network, but their roles have not been widely experimentally evaluated. In this study, the small number of cells exhibited activity and stable currents in the NavAb M181A mutants, particularly in S178T^NK/TA-M181A. Met181 of NavAb is suggested to promote monovalent cation selectivity through SF stabilization, and this effect is more pronounced in the S178T mutant. This combination effect of Met181 and S178T may be a reason for the increasing trend (though without a significant difference from N49K/T206A) in Li⁺ selectivity in S178T^NK/TA (Fig. 2 E and Table 1), which retains the first hydration water exchange site. To better understand the molecular basis of the Li⁺ selectivity of Nav, detailed analyses of these extensive interactions are required, especially those between the SF and pore helices. The specific roles of Gln172 and Met181 could not be clarified, despite being strong candidates. Further mutational analyses are thought to be required to elucidate the functions of these residues.

### NavAb Ser178 mutants are a clue to understanding the mechanism of Li⁺ selectivity of eukaryotic Navs

Our results suggest mechanisms for Li⁺ selectivity in a prokaryotic Nav and indicate that similar mechanisms may operate in eukaryotic Navs, which evolved from prokaryotic channels. Unlike prokaryotic Navs, eukaryotic Navs are heteromeric and exhibit sequence variation among subtypes. Regarding the human Nav subtypes 1.1–1.9 targeted by lithium preparations, there are four distinct residue patterns at the position corresponding to NavAb Ser178. Specifically, the compositions of domains from I to IV are F–GG (Nav1.1, 1.2), Y–GG (Nav1.3, 1.4, 1.6, 1.7), C–GG (Nav1.5), and S–GG (Nav1.8, 1.9), where the hyphen (-) indicates the absence of the corresponding residue (Li et al., 2024). Similar to NavAb S178G, the deletion in domains from II to IV and the presence of glycine residues may widen the SF entrance, potentially contributing to the Li⁺ selectivity observed in Navs of higher animals. Consequently, as next steps, molecular dynamics simulations of NavAb with a heteromeric SF configuration, combined with further mutational analysis of residues surrounding the SF, will be important in elucidating the molecular basis of the high Li⁺ selectivity in eukaryotic Navs.

Furthermore, in this study, we gave particular focus to factors contributing to the increased $P_{Li}/P_{Na}$ ratios with attention to changes in hydration water exchange in the SF and structural alterations in NavAb Ser178 mutants. Importantly, $P_{Li}/P_{Na}$ ratio represents the relative permeability of each ion, but it is unclear whether its increase reflects enhanced Li⁺ permeability or reduced Na⁺ permeability. The precise ion permeability of ion channels is evaluated by measuring the channel conductance through single-channel recordings, which is suitable for channels that spontaneously open without voltage sensors. Single-channel recordings have been achieved using homologs of BacNavs with removed voltage sensors (Shaya et al., 2011), so creating similar Li⁺-selective mutants in these homologs would likely enable more detailed evaluation.

In summary, we analyzed the mechanisms of high Li⁺ selectivity in Navs using a BacNav. It was suggested that the strong electrostatic interaction in the SF was the main contributor to

the high Li⁺ selectivity. Also, fewer hydration water exchanges promote Li⁺ permeation in Navs. Differences in ionic properties, such as ionic radius, electrostatic force, and hydration water exchange rate, may be related to the Li⁺ selectivity of Navs. We also revealed that the SF stabilization by the extracellular funnel and pore helices supports the selective permeation of monovalent cations. These discoveries of the molecular basis of the high Li⁺ selectivity of Navs and the creation of LivAb, a novel "Li channel," provide direction for drug development targeting various neurological disorders. This may help in the understanding of the biological effects of lithium.

## Data availability
The data that support this study are available from the corresponding authors upon reasonable request. The structural data generated in this study have been deposited in the Protein Data Bank under accession codes 8H9W (NavAb N49K mutant in calcium), 9UC1 (NavAb S178T$^{NK}$ mutant), 9UC2 (NavAb S178A$^{NK}$ mutant), 9UC3 (NavAb S178G$^{NK}$ mutant [LivAb$^{NK}$]), and 9UC4 (NavAb S178T$^{NK/TA}$ mutant). The structure of the NavAb N49K mutant for the initial model of molecular replacement was available in the Protein Data Bank under accession code 8H9W.

## Acknowledgments
Christopher J. Lingle served as editor.

The synchrotron radiation experiments were performed at BL41XU and BL32XU in SPring-8 with the approval of the Japan Synchrotron Radiation Research Institute (proposal numbers 2016B2721, 2017B2735, and 2018B2710). We thank the beamline staff for their excellent facilities and support. We acknowledge proofreading and editing by Benjamin Phillis, a Board-Certified Editor in the Life Sciences, at Wakayama Medical University.

This research was partially supported by Platform Project for Supporting Drug Discovery and Life Science Research (Basis for Supporting Innovative Drug Discovery and Life Science Research) from the Japan Agency for Medical Research and Development under grant numbers JP22ama121001 and JP23ama121001. This work was supported by Grants-in-Aid for Scientific Research (17K17795, 20K09193, and 24K02168) and SEI Group CSR Foundation, Takeda Science Foundation, Institute for Fermentation (G-2021-2-020), and the Salt Science Research Foundation (202403).

Author contributions: Yuki K. Maeda: conceptualization, data curation, formal analysis, investigation, methodology, resources, software, validation, visualization, and writing—original draft, review, and editing. Kentaro Kojima: investigation and writing—review and editing. Tomoe Y. Nakamura: writing—review and editing. Toru Nakatsu: validation and writing—review and editing. Katsumasa Irie: conceptualization, data curation, formal analysis, funding acquisition, investigation, methodology, project administration, resources, software, supervision, validation, visualization, and writing—original draft, review, and editing.

Disclosures: The authors declare no competing interests exist.

Submitted: 17 July 2025

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

# Supplemental material

**Provided online is Table S1. Table S1 shows data collection and refinement statistics.**

