## [Peer Review File · The Journal of General Physiology]

Structure-function analysis of the lithium-ion selectivity of the voltage-gated sodium channel

Yuki Maeda, Kentaro Kojima, Tomoe Nakamura, Toru Nakatsu, and Katsumasa Irie

Corresponding Author(s): Katsumasa Irie, Wakayama Medical University

Review Timeline:

Submission Date:	July 17, 2025
Editorial Decision:	August 26, 2025
Revision Received:	December 19, 2025
Editorial Decision:	January 16, 2026
Revision Received:	January 28, 2026
Editorial Decision:	January 28, 2026
Revision Received:	February 3, 2026

Editor: Christopher Lingle

Transaction Report:

DOI: <https://doi.org/10.1085/jgp.202513855>

August 26, 2025

Dr. Katsumasa Irie
Wakayama Medical University
25-1, Shichibancho
Wakayama, Wakayama 640-8156
Japan

Re: 202513855

Dear Dr. Irie,

Thank you for submitting your manuscript, entitled "Structure-function analysis of lithium-ion selectivity of voltage-gated sodium channel" to JGP. Your manuscript has now been seen by 3 reviewers, whose comments are appended below. You will see that the reviewers were quite positive about the study and its potential impact. Overall, the concerns were generally minor, but will nevertheless need to be addressed prior to further consideration of the manuscript at JGP. Below I summarize some of the main concerns that were raised, but be sure to address all the points mentioned by the reviewers in your revisions.

1. Reviewers noted that important experimental details were in some cases omitted. This includes any necessary series resistance compensation, impact of junction potential measurements on estimates of reversal potential, and impact of any leak currents on reversal potential estimates. Is the -8 mV apparent reversal potential for Li/Na for the WT background construct due to junctional potential issues or a true difference in selectivity? With the methods employed in this paper, do Na/Na gradients yield a 0 mV reversal?
 2. Revs. #2 and #3 both pointed out that any conclusions regarding the basis for the differences in selectivity and potential role of differences in hydration water exchanges is quite speculative. This should be treated as such in your presentation. Rev. #3 suggested that additional discussion on this topic may be warranted. However, in lieu of additional evaluations (e.g., perhaps MD simulations, structures in Li⁺), it is difficult to see how additional insight into the basis for the selectivity differences. However, the reviewers (and the Associate Editors) still consider this work of value in guiding future work on this topic.
 3. A question was raised among the Associate Editors regarding the statistical information or lack of it in regards to some of the results. For example, Table 1 lacks any statistical comparisons among constructs for PLi/PNa. We note that Table 2 does in fact include P values. Per JGP policy, we prefer that exact P values be provided, even when NS.
- Also, per the policy of JGP, we would recommend, for the data in Figure 2 and in some other comparisons of constructs that follow (e.g., Fig. 3 and Fig. 7), that panels be added of bar plots that show, for the set of reversal potential measurements, the mean+SD and individual values for reversal potentials for each of the cells used to determine the average for a given construct.
4. In the discussion of your manuscript among the Associate Editors, question was raised whether the change in selectivity in the S178 mutants might reflect a relative decrease in Na permeability, rather than an increase in Li permeability.

We hope that you will be able to submit a revised manuscript that addresses these points, which we believe will pose no problems, and which may be re-reviewed. In addition, please do not hesitate to contact me (via the editorial office) if you feel that a discussion of the reviewers' and editors' comments would be helpful.

Please submit your revised manuscript via the link below, along with a point-by-point letter that details your response to the reviewers' and editor's summary, as well as a copy of the text with alterations highlighted (boldfaced or underlined). If the article is eventually accepted, it would include a 'revised date' as well as submitted and accepted dates. If we do not receive the revised manuscript within one year, we will regard the article as having been withdrawn. We would be willing to receive a revision of the manuscript at a later time, but the manuscript will then be treated as a new submission, with a new manuscript number.

Please pay particular attention to recent changes to our instructions to authors in the following sections: Data presentation, Blinding and randomization and Statistical analysis, under Materials and Methods, as shown here: <https://rupress.org/jgp/pages/submission-guidelines#prepare>. Re-review will be contingent on inclusion of the required information (including for data added during revision) and demonstration of the experimental reproducibility of the results. Also, To improve the reproducibility of published content, we have partnered with SciScore. Authors are prompted in eJP to copy and paste the Materials and Methods section of their manuscript for a SciScore assessment when submitting their revised manuscript. Authors are encouraged (not required) to further revise their Materials and Methods if the SciScore is below 4. More information can be found here: <https://rupress.org/jgp/pages/submission-guidelines#sciscore>.

Please note, JGP now requires authors to submit Source Data used to generate figures containing gels and Western blots with all revised manuscripts (when applicable). This Source Data consists of fully uncropped and unprocessed images for each gel/blot displayed in the main and supplemental figures. If your paper includes cropped gel and/or blot images, please be sure to provide one Source Data file for each figure that contains gels and/or blots along with your revised manuscript files. File names for Source Data figures should be alphanumeric without any spaces or special characters (i.e., SourceDataF#, where F# refers to the associated main figure number or SourceDataFS# for those associated with Supplementary figures). The lanes of the gels/blots should be labeled as they are in the associated figure, the place where cropping was applied should be marked (with a box), and molecular weight/size standards should be labeled wherever possible.

Source Data files will be made available to reviewers during evaluation of revised manuscripts and, if your paper is eventually published in JGP, the files will be directly linked to specific figures in the published article.

Source Data Figures should be provided as individual PDF files (one file per figure). Authors should endeavor to retain a minimum resolution of 300 dpi or pixels per inch. Please review our instructions for export from Photoshop, Illustrator, and PowerPoint here: <https://rupress.org/jgp/pages/submission-guidelines#revised>

Whilst you are revising your manuscript, we ask that you consider whether you have any artwork that might be suitable for the cover of JGP. Microscopy images are particularly good for cover artwork, but other types of image can be very effective, so we encourage you to be creative. Please don't restrict yourself to images from the paper; an image that is relevant to the work described would be just as suitable. Images should be a minimum resolution of 300 dpi. To see recent examples, visit the following page and click on 'Show covers? Yes': <https://jgp.rupress.org/content/by/year>

Thank you for submitting your interesting research to JGP.

Please submit your revised manuscript, and any associated files, via this link:

Link Not Available

Sincerely,

Christopher Lingle, Ph.D.

On behalf of Journal of General Physiology

Journal of General Physiology's mission is to publish mechanistic and quantitative molecular and cellular physiology of the highest quality; to provide a best-in-class author experience; and to nurture future generations of independent researchers.

Reviewer #1 (Comments to the Authors):

In this paper, the authors aimed at elucidating the molecular mechanism of selective permeation of Li⁺ through Nav channel. They utilized the 1 repeat- type homo- tetrameric bacterial Na⁺ channel (NavAb) which this group has been working on.

They analyzed electro physiologically the function of some mutants including T178G and M181A, and also solved their crystal structure of the T178 mutants.

They observed that Li⁺ permeability relative to Na⁺ of T178G is significantly elevated. They showed in T178G that the number of hydration exchange site is less, that the strong electrostatic site, Glu177, is better exposed, and that the SF vestibule is widened facilitating the approach of bulk solution. Based on these data, they concluded that the smaller number of hydration exchanges and the strong electrostatic force in the SF mainly contribute to the high Li⁺ selectivity.

They also observed electro physiologically that the Ca²⁺ selectivity is elevated by adding M181A mutation. They discussed that Met181 promotes monovalent cation selectivity (less Ca²⁺ selectivity) through SF stabilization.

Overall, this study is thoroughly done and scientifically sound with no major flaws. Also, the findings of Li⁺ selective channel following Eisenmann Seq XI have a high impact to the readers of ion channel research field. The well written introduction, explaining the significance of Li permeation in physiological and pathophysiological conditions, will attract broader ranged readers.

To elevate the quality of the paper, I described in the following some mostly minor comments which require attention.

Major comments

[1] In the introduction, the authors started with the detailed explanation about the significance of Li⁺ permeation in Eukaryotic (mammalian) Nav, but nothing is described in the discussion section. I wonder if the findings on NavAb in the present study

would also apply to Nav.

(1) What are the 4 amino acid residues in the 4 repeats of Nav, corresponding to Thr178?

(2) Is it practically possible to analyze the Li⁺ permeation in the mammalian Nav quadra mutants corresponding to NavAb T178G (with a slow inactivation mutation)? At the least, I would like to have some discussion about this point.

[2] Proteins were crystalized in 100 mM NaCl and other ions. What will happen to the structure, if crystalized in 100 mM Li⁺ instead of 100 mM Na⁺? Will there be any change of the protein structure due to a sort of induced fit? I would like to have the opinion/discussion of the authors.

Minor comments

[1] Line 110: NavAb: There is no explanation what this is.

[2] Line 115: S178T NK: There is no explanation about NK. The explanation about NK in lines 116-118 should be moved to line 114. Also, where does N49 locate in the protein?

[3] Figure 1: In Figure 2 there is a description about the solutions. Here the intracellular solution is not described.

[4] Line 131: Where does T206 locate in the protein? Although it is written in line 159 that "T206 is located in the channel lumen", it should be described here.

[5] Figure 2 and related text: It is better to state here that the leak current subtraction was done online, although it is described in the method section.

[6] Figure 3: Indication of extracellular solutions, K⁺, Cs⁺, Ca²⁺ is too small and hard to read. It is better to indicate at the whole top.

[7] Figure 4: An explanation about the green particle is missing.

[8] Figure 4 and related text: Isn't is necessary to briefly describe here that the protein was crystalized in Na⁺, not in Li⁺, although the detail is described in the method.

[9] Figure 6: "Arg185 interaction was lost in S178G NK". Interaction with what? Does it mean interaction between Arg185 and Met181? I cannot find clear differences from other mutants. Also, I cannot understand the relationship with the description in lines 206-210, "similar interactions were observed in all mutants"

[10] Discussion: Relevant figures could be cited in each part to help readers' understanding.

[11] Method, line 371: There is no description about the error of the clamped voltage due to series resistance, and its compensation. How about the resistance of the patch pipette? How and to which extent was series resistance compensated? For example, in Figure 2B right panel, the amplitude of the inward current is over 10 nA, and the error is obviously very serious. Please justify that the error does not significantly affect the evaluation of the reversal potentials (zero current membrane potential) which are most critical in this study.

Reviewer #2 (Comments to the Authors):

This manuscript investigates the defining characteristic of sodium channels, the selective permeation of Na⁺. There are a number of structures reported for Na⁺ channels. These studies have clearly defined the architecture of the selectivity filter, the ion binding sites in the channel. Many mutational studies on the ion binding sites have also been carried out. Even with these structural and functional studies, the mechanistic basis for the selective permeation of Na⁺ is not yet firmly established. Here the authors focus on a Ser residue, Ser178, in the selectivity filter sequence. They carry out substitutions at this Ser residue and show that the Ser178Gly substitution shows a greater selectivity for Li compared to Na⁺. The mutant does not alter the ability of the channel to discriminate between Na⁺ and K⁺, Cs⁺ or Ca²⁺. This indicates a clean switch in selectivity of the channel from Na⁺ to Li⁺. The authors report the crystal structures of the Ser179 mutants, which were determined to a moderate resolution. The structures of the Ser179 mutants are basically identical to the wild type channel. These structural and functional studies of the Li⁺ selective variant are the strengths of the manuscript.

Ser178 is at the mouth of the channel pore and the authors suggest that the lack of the side chain in the Ser178Gly mutant leads to an increase in water molecules that are present around the mouth. The authors make this claim based on the structural data, but the evidence presented is not very convincing as the water electron density observed is weak. The authors need to provide more convincing evidence for why water was modelled at these weak puffs of electron density seen. However, a potential change in the hydration at the mouth of the pore and the idea that there are fewer sites at which the water molecules hydrating the permeating Li⁺ ion are exchanged for channel ligands, is an interesting idea for why the mutant channel may have changed from being Na⁺ selective to Li⁺ selective. This is however a hypothesis and should be clearly defined as such and is

more appropriate in the discussion and not in the results. The additional mutational data presented in the manuscript on M181A and Q127N do not provide any additional clarity.

The major strength of the manuscript is the structure of a channel with selectivity switched from Na⁺ to Li⁺. The structure data presented will be useful for computational studies on the mechanism of ionic selectivity in Na⁺ channels.

Minor points:

The authors claim that Li⁺ has a stronger positive charge than Na⁺ (Line:306). It should be changed to say that Li⁺ has a higher charge density than Na⁺.

An Fo-Fc omit map with the residue omitted should be used to show the fit of the side chain to the electron density.

Reviewer #3 (Comments to the Authors):

In this work, Madea et al. report that mutations of S178 in NavAb enhanced Li-selectivity in resulting channels, with S178G converting NavAb into a Li-selective channel. The authors also solved structures of NavAb S178 mutants, suggesting that S178 mutations remove the first hydration water exchange site in the selectivity filter (SF) of NavAb to enhance Li-selectivity by three mechanisms: reducing the number of hydration water exchanges during Li⁺ permeation, allowing stronger electrostatic interaction between hydrated Li⁺ and Glu177 in the SF, and altering the supportive network behind the SF. The Li-selective NavAb constructs developed in this work could serve as an important tool to study Li⁺-selectivity in Nav channels. However, the differences between the structures of WT and S178-mutated channels are quite subtle, which makes some conclusions in the paper less well-founded. Below are my specific comments.

1. The reversal potential was determined by a voltage ramp protocol in this study. It may be necessary to verify the results with instantaneous I-V recordings, particularly in cases where the reversal potential lies within the deactivation voltage range. Alternatively, the reversal potential could be measured with Na on the extracellular side and less-permeable cations on the intracellular side, ensuring the reversal potential is within the activation voltage range of NavAb.
2. How were the series resistance error and liquid junction potential compensated in the experiment?
3. It was mentioned in the Methods that "Cells with a smaller leak current than 1nA were used for data collection" (page 21, line 401). Considering the amplitude of sample currents shown in the figures, a threshold of 1nA for leak current seems excessively large.
4. One major conclusion from this work is that Li⁺ selectivity is enhanced by removing the first water exchange site and widening of extracellular entrance of SF in NavAb. However, the results of S178T (increased Li⁺ selectivity, seemingly unchanged water exchange sites, and narrower extracellular entrance) appear to contradict other conclusions. The authors should include a more detailed discussion about this.
5. Could the main chain oxygens of S178G form a water exchange site in the mutant?
6. In Fig. 6, it is difficult to discern how the network involving M181 in the S178G structure differs from that in other NavAb constructs. Even if the difference in M181 observed in the S178G structure is valid, the functional result that M181A only changes the Ca²⁺-selectivity of NavAb does not support authors' statement that "The difference in Met181 observed in S178G may, therefore, be responsible for improving Li selectivity" (page 12 line 211).
7. The differences among the structures of S178 mutants are quite subtle. This is not unexpected given that the structures of these NavAb constructs with similar selectivity for Na were determined in NaCl. Is it possible to solve the structures in LiCl?
8. The quality of the English writing in this paper could be improved, and seeking professional editorial assistance might be beneficial.

We sincerely appreciate the editor and reviewers' valuable and insightful comments on
our manuscript. Our point-by-point responses to *each reviewer's comments (italic text)*
are listed below. The textural changes in the revised manuscript are shown in the
underlined text. All the line numbers indicated in this rebuttal are referred to in the
revised version.

***Editor comments***

*Thank you for submitting your manuscript, entitled "Structure-function analysis of*
*lithium-ion selectivity of voltage-gated sodium channel" to JGP. Your manuscript has*
*now been seen by 3 reviewers, whose comments are appended below. You will see that*
*the reviewers were quite positive about the study and its potential impact. Overall, the*
*concerns were generally minor, but will nevertheless need to be addressed prior to*
*further consideration of the manuscript at JGP. Below I summarize some of the main*
*concerns that were raised, but be sure to address all the points mentioned by the*
*reviewers in your revisions*

*1. Reviewers noted that important experimental details were in some cases omitted. This*
*includes any necessary series resistance compensation, impact of junction potential*
*measurements on estimates of reversal potential, and impact of any leak currents on*
*reversal potential estimates. Is the -8 mV apparent reversal potential for Li/Na for the*
*WT background construct due to junctional potential issues or a true difference in*
*selectivity? With the methods employed in this paper, do Na/Na gradients yield a 0 mV*
*reversal?*

Thank you for your comments on the electrophysiological recording conditions. As
you noted, proper control of series resistance, liquid junction potential, and leak current
are essential for accurately measuring reversal potentials. Accordingly, we have added
details of these procedures to the *Materials and Methods* section.

In this study, the liquid junction potential was compensated to zero before seal
formation. We also prepared an external solution containing 150 mM Na⁺, identical to
the monovalent cation solution used in other experiments. We measured the reversal
potential under Na⁺/Na⁺ conditions for each construct (NavAb N49K/T206A,
S178T^{NK/TA}, S178A^{NK/TA}, and S178G^{NK/TA}). The presence of 2 mM Ca²⁺ in the external
solution, which is used to stabilize the seal, caused a slight positive shift (+4.0 to +5.2
34 mV) in the reversal potential from zero mV, but the deviation remained within expected
values. The mean deviation of the reversal potential from zero mV in the condition of

150 mM $[\text{Na}^+]_{\text{out}}$, 2 mM $[\text{Ca}^{2+}]_{\text{out}}$ and 150 mM $[\text{Na}^+]_{\text{in}}$ was follows: +3.98 mV, +5.0 mV,
4.6 mV, and 5.2 mV for N49K/T206A (n=4), S178T^{NK/TA} (n=3), S178A^{NK/TA} (n=4), and
S178G^{NK/TA} (n=2), respectively. Importantly, reversal potentials were consistent among
constructs, indicating that differences observed under the external solutions containing
Li^+ , K^+ , Cs^+ , and Ca^{2+} reflect true variations in ion selectivity. This information enhances
the accuracy of the ion selectivity analysis in this study, and we have incorporated it into
the *Results* section. We appreciate your constructive comments, which have helped
strengthen the manuscript.

The following text was added with the above points in mind:

Section: **Electrophysiological measurement in insect cells**

(Page 16, Line 474)

“Whole-cell recordings were obtained using patch pipettes with resistances ranging
from 1.9 to 5.0 M Ω .”

(Page 16, Line 486)

“The pipette current was zeroed before forming a seal with each cell to account for the
2.25 mV liquid junction potential. Series resistance was compensated and maintained
below 10 M Ω before each current recording. Cells with leak current < 800 pA were
used for data collection, and the mean leak current ranged from 110 to 190 pA for each
construct.”

Section: **Electrophysiological evaluation of ion selectivity**

(Page 7, Line 151)

“Before evaluating ion selectivity, we measured reversal potentials under iso-ionic
conditions for each construct using solutions containing 150 mM $[\text{Na}^+]_{\text{out}}$ and 150 mM
$[\text{Na}^+]_{\text{in}}$ (Figure 2B). The reversal potential was +3.98 to +5.20 mV in N49K/T206A,
S178T^{NK/TA}, S178A^{NK/TA}, and S178G^{NK/TA}, which was slightly positive due to the
presence of 2 mM Ca^{2+} in the external solution. To stabilize the measurement, Ca^{2+}
ions were added to the external solution. The positive deviation from zero mV caused
by Ca^{2+} ions remained within one order of magnitude mV, so we considered that 2 mM
Ca^{2+} did not disturb the evaluation of the selectivity of other ions.”

*2. Revs. #2 and #3 both pointed out that any conclusions regarding the basis for the*
*differences in selectivity and potential role of differences in hydration water exchanges*
*is quite speculative. This should be treated as such in your presentation. Rev. #3*
*suggested that additional discussion on this topic may be warranted. However, in lieu of*

additional evaluations (e.g., perhaps MD simulations, structures in Li^+), it is difficult to
see how additional insight into the basis for the selectivity differences. However, the
reviewers (and the Associate Editors) still consider this work of value in guiding future
work on this topic.

Thank you for your comments. As you rightly pointed out, the conclusions regarding
the mechanism underlying enhanced Li^+ selectivity in this study remain speculative. In
line with the comments, particularly the comment of Reviewer #3 regarding the S178T
mutant, we performed additional statistical analyses to examine further the difference in
Li^+ selectivity of the Ser178 mutants, thereby deepening our interpretation of the results.
Dunnett's test revealed significant differences in Li^+ selectivity for the S178G and
S178A mutants compared with N49K/T206A, but not for S178T. Although the trend
toward increased Li^+ selectivity in S178T warrants further investigation, these statistical
results support our proposed mechanism for Li^+ selectivity, regarding hydration water
exchanges. In the revised manuscript, we have also refined the wording and sentence
structure to ensure a more apparent distinction between the Results and Discussion
sections.

Also, as you also pointed out, structural data from crystal structures or molecular
dynamics simulations incorporating Li^+ would provide essential insights into the
structural basis of Li^+ selectivity. However, crystal quality deteriorated markedly in the
presence of Li^+ , preventing us from obtaining structural data for the Li^+ -containing
NavAb mutants. Furthermore, molecular dynamics simulations still face challenges in
accurately reproducing the behavior of ions with strong electrostatic forces, such as Li^+
and Ca^{2+} , which remains an important task for our future research. Although much
remains to be elucidated regarding the molecular basis of Li^+ selectivity in Navs, this
study represents the first proposal to address this issue. We sincerely appreciate your
understanding of this point. In response to these points about the results of Dunnett's
test and the clarification of results and discussion regarding the mechanisms of
Li^+ -selectivity, the manuscript has been modified as follows:

Section: **Electrophysiological evaluation of ion selectivity**

(Page 7, Line 166)

“S178A^{NK/TA} also showed a positive shift of reversal potentials relative to N49K/T206A,
but to a lesser extent than S178G^{NK/TA}. A slight positive shift was also shown by
S178T^{NK/TA}, but there was no significant difference of $P_{\text{Li}}/P_{\text{Na}}$ compared to
N49K/T206A (Figure 2E).”

(Page 8, Line 195)

“To obtain high-resolution diffraction patterns, crystallization in the presence of Na⁺
and diffraction experiments in high-Na⁺ solutions were essential. Unfortunately, we
were unable to obtain protein crystals under Li⁺ conditions without Na⁺, and when the
crystals obtained under Na⁺ conditions were transferred to a Li⁺-containing solution, the
diffraction patterns disappeared in the diffraction experiments.”

(Page 9, Line 225)

“The smaller number of hydration water exchange sites would be a candidate for the Li⁺
selectivity promoter. At the same time, smaller side-chain mutations exposed the
negatively charged side chain of Glu177, and this exposed and negatively charged side
chain would also be a candidate for the Li⁺ selectivity promoter.”

(Page 9, Line 238)

“On the other hand, this is inconsistent with S178T^{NK} (which has narrower pores)
showing a slight improvement in Li⁺ selectivity, although without significant difference.

“

(Page 12, Line 346)

“In NavAb S178A^{NK/TA} and S178G^{NK/TA} (LivAb^{NK/TA}) mutants, which had greatly
improved Li⁺ selectivity compared to N49K/T206A (Figure 2E and Table 1), the loss of
the hydration water exchange site formed by the side chain of Ser178 would expose
Glu177 directly to the bulk solution (Figures 6C, D and 11).”

(Page 12, Line 360)

“If the vicinal electron densities were water molecules, it would explain the high
accessibility of hydrated ions in bulk solution to the most electrostatic sites in these
mutants.”

(Page 13, Line 399)

“This combination effect of Met181 and S178T may be a reason for the increasing trend
(though without significant difference from N49K/T206A) in Li⁺ selectivity in
S178T^{NK/TA} (Figure 2E and Table 1), which retains the first hydration water exchange
site.”

(Page 14, Line 436)

“In summary, we analyzed the mechanisms of high Li⁺ selectivity in Navs using a
BacNav. It was suggested that the strong electrostatic force in the SF was the main
contributor to the high Li⁺ selectivity. Also, fewer hydration exchanges promote Li⁺
permeation in Navs.”

*3. A question was raised among the Associate Editors regarding the statistical*
*information or lack of it in regards to some of the results. For example, Table 1 lacks*

any statistical comparisons among constructs for P_{Li}/P_{Na} . We note that Table 2 does in
fact include *P* values. Per JGP policy, we prefer that exact *P* values be provided, even
when NS.

Also, per the policy of JGP, we would recommend, for the data in Figure 2 and in some
other comparisons of constructs that follow (e.g., Fig. 3 and Fig. 7), that panels be
added of bar plots that show, for the set of reversal potential measurements, the
mean+SD and individual values for reversal potentials for each of the cells used to
determine the average for a given construct.

Thank you for your comments regarding the statistical analysis. In accordance with the
JGP guidelines, we have added supplementary tables (Supplementary Tables 1 and 2)
presenting the statistical results corresponding to Tables 1 and 2, which summarize the
ion selectivity analyses. These results were obtained using Student's *t*-test and the newly
performed Dunnett's test. In addition, we have included bar graphs showing the ion
selectivity values, the mean \pm SE for each cell, and the statistical results (Figures 2E, 4A,
4B, 9B, 9C and 10 in the updated manuscript).

4. In the discussion of your manuscript among the Associate Editors, question was
raised whether the change in selectivity in the S178 mutants might reflect a relative
decrease in Na permeability, rather than an increase in Li permeability.

Thank you for your comment. It is indeed unclear whether the increase in PLi/PNa
reflects enhanced Li⁺ permeability or reduced Na⁺ permeability. In this study, based on
the loss of a hydration water exchange site at the entrance of the selectivity filter and the
electron density changes observed in NavAb S178G mutant, we hypothesized that the
increase of PLi/PNa arises from enhanced Li⁺ permeability. To conclusively determine
whether the change is due to increased Li⁺ or decreased Na⁺ permeability, we plan to
evaluate ion permeability using single-channel recordings. However, because NavAb
has voltage dependence, it cannot be directly used for single-channel recordings. We
therefore intend to perform experiments using constructs that have the selectivity filter
of NavAb without the voltage sensor. As conducting such experiments is still ongoing,
we have submitted this paper to offer new hypotheses about the mechanisms of Li
selectivity in Navs. Based on your feedback, we have added the following text to
*Discussion* section. Thank you for the valuable feedback.

Section: **NavAb Ser178 mutants are clue to understand the mechanism of Li⁺**
**selectivity of eukaryotic Navs**

(Page 14, Line 425)

“Furthermore, in this study, we gave particular focus to factors contributing to the
increased Li^+ permeability with attention to changes in hydration water exchange in
the SF and structural alterations in NavAb Ser178 mutants. Importantly, $P_{\text{Li}}/P_{\text{Na}}$
represent the relative permeability of each ion, but it is unclear whether their increase
reflects enhanced Li^+ permeability or reduced Na^+ permeability. The precise ion
permeability of ion channels is evaluated by measuring the channel conductance
through single-channel recordings, which is suitable for channels that spontaneously
open without voltage sensors. Single-channel recordings have been achieved using
homologs of BacNavs with removed voltage sensors (Shaya et al., 2011), so creating
similar Li^+ -selective mutants in these homologs would likely enable more detailed
evaluation.”

**Reviewer #1**

*In this paper, the authors aimed at elucidating the molecular mechanism of selective*
*permeation of Li⁺ through Nav channel. They utilized the 1 repeat- type homo-*
*tetrameric bacterial Na⁺ channel (NavAb) which this group has been working on.*

*They analyzed electro physiologically the function of some mutants including T178G*
*and M181A, and also solved their crystal structure of the T178 mutants.*

*They observed that Li⁺ permeability relative to Na⁺ of T178G is significantly elevated.*
*They showed in T178G that the number of hydration exchange site is less, that the*
*strong electrostatic site, Glu177, is better exposed, and that the SF vestibule is widened*
*facilitating the approach of bulk solution. Based on these data, they concluded that the*
*smaller number of hydration exchanges and the strong electrostatic force in the SF*
*mainly contribute to the high Li⁺ selectivity.*

*They also observed electro physiologically that the Ca²⁺ selectivity is elevated by*
*adding M181A mutation. They discussed that Met181 promotes monovalent cation*
*selectivity (less Ca²⁺ selectivity) through SF stabilization.*

*Overall, this study is thoroughly done and scientifically sound with no major flaws.*
*Also, the findings of Li⁺ selective channel following Eisenmann Seq XI have a high*
*impact to the readers of ion channel research field. The well written introduction,*
*explaining the significance of Li permeation in physiological and pathophysiological*
*conditions, will attract broader ranged readers.*

*To elevate the quality of the paper, I described in the following some mostly minor*
*comments which require attention.*

*Major comments*

*[1] In the introduction, the authors started with the detailed explanation about the*
*significance of Li⁺ permeation in Eukaryotic (mammalian) Nav, but nothing is described*
*in the discussion section. I wonder if the findings on NavAb in the present study would*
*also apply to Nav.*

*(1) What are the 4 amino acid residues in the 4 repeats of Nav, corresponding to*
*Thr178?*

*(2) Is it practically possible to analyze the Li⁺ permeation in the mammalian Nav*
*quadra mutants corresponding to NavAb T178G (with a slow inactivation mutation)? At*
*the least, I would like to have some discussion about this point.*

**Reply to [1]:**

Thank you for your thoughtful comments. Regarding your point, it remains unclear

whether the findings on NavAb presented in this study can be directly applied to Navs
in higher animals. Nevertheless, our work provides the first insight into the molecular
basis of Li⁺ selectivity in Navs, and we hope it will serve as a foundation for future
studies to further explore this molecular basis in eukaryotic Navs.

As you are aware, the amino acid residues corresponding to Glu177 of NavAb in the
selectivity filter of eukaryotic Navs are known to be critical for Na⁺ selectivity.
However, the heteromeric nature of their selectivity filter makes it difficult to determine
which residues contribute to ion permeability and in what manner. Therefore, we
believe that reproducing the asymmetric selectivity filter and Li⁺ permeability of NavAb
through molecular dynamics simulations represents a feasible approach to elucidating
Li⁺ selectivity in eukaryotic Navs.

**Reply to (1):** Thank you for your question. Regarding the hNav subtypes 1.1–1.9
targeted by lithium preparations, there are four distinct residue patterns at the position
we focused on in this study. Specifically, the compositions of domains from I to IV are
F–GG (Nav1.1, 1.2), Y–GG (Nav1.3, 1.4, 1.6, 1.7), C–GG (Nav1.5), and S–GG (Nav1.8,
1.9), where the hyphen (-) indicates the absence of the corresponding residue. Although
the residue in domain I varies among F, Y, C, and S, domains II–IV consistently share
the –GG motif. Similar to NavAb S178G, the deletion in domains II–IV and the
presence of glycine residues may widen the SF entrance, potentially contributing to the
Li⁺ selectivity observed in Navs of higher animals.

**Reply to (2):** Thank you for your question. To state our conclusion first, we believe it is
currently challenging to analyze the structural basis of Li⁺ selectivity of mammalian
Navs using experiments similar to those conducted in this study. The primary reason is
that, unlike prokaryotic Navs, mammalian Navs are heteromeric. Their high ion
selectivity arises from a complex interplay among ions, water molecules, and residues
within the heteromeric selectivity filter, including the exchange of hydration water.
Moreover, even if a mutation at the residue corresponding to NavAb Ser178 were to
alter Li⁺ selectivity, achieving high-resolution structural analysis remains technically
challenging at present. For these reasons, we still consider analysis of the structural
basis of Li⁺ selectivity in mammalian Navs through similar experimental approaches to
be challenging.

NavAb, on the other hand, is a channel whose Na⁺ permeation behavior has been
successfully reproduced through molecular dynamics simulations. We therefore believe
that the next feasible step for more detailed analysis would be reconstructing the

heteromeric selectivity filter of mammalian Navs using NavAb as a structural model,
and simulating Na⁺ and Li⁺ permeation. In parallel with continued electrophysiological
investigations of residues around the P-helices, such as NavAb Met181 and Gln172, the
most direct route to elucidating the mechanism of Li⁺ selectivity in mammalian Navs
may be by clarifying the structural basis of Li⁺ selectivity in NavAb, where structural
determination is achievable.

In response to your sharp perspective in comment [1], we have added the following
text to the *Discussion* section:

Section: **NavAb Ser178 mutants are clue to understanding the mechanism of Li⁺**
**selectivity of eukaryotic Navs**

(Page 14, Line 410)

“Our results suggest mechanisms for Li⁺ selectivity in a prokaryotic Nav and indicate
that similar mechanisms may operate in eukaryotic Navs, which evolved from
prokaryotic channels. Unlike prokaryotic Navs, eukaryotic Navs are heteromeric and
exhibit sequence variation among subtypes. Regarding the human Nav subtypes 1.1–1.9
targeted by lithium preparations, there are four distinct residue patterns at the position
corresponding to NavAb Ser178. Specifically, the compositions of domains from I to IV
are F–GG (Nav1.1, 1.2), Y–GG (Nav1.3, 1.4, 1.6, 1.7), C–GG (Nav1.5), and S–GG
(Nav1.8, 1.9), where the hyphen (-) indicates the absence of the corresponding residue
(Li et al., 2024). Similar to NavAb S178G, the deletion in domains from II to IV and the
presence of glycine residues may widen the SF entrance, potentially contributing to the
Li⁺ selectivity observed in Navs of higher animals. Consequently, as next steps,
molecular dynamics simulations of NavAb with a heteromeric SF configuration,
combined with further mutational analysis of residues surrounding the SF, will be
important in elucidating the molecular basis of the high Li⁺ selectivity in eukaryotic
Navs.” “Furthermore, in this study, we gave particular focus to factors contributing to
the increased Li⁺ permeability with attention to changes in hydration water exchange in
the SF and structural alterations in NavAb Ser178 mutants. Importantly, P_{Li}/P_{Na}
represent the relative permeability of each ion, but it is unclear whether their increase
reflects enhanced Li⁺ permeability or reduced Na⁺ permeability. The precise ion
permeability of ion channels is evaluated by measuring the channel conductance
through single-channel recordings, which is suitable for channels that spontaneously
open without voltage sensors. Single-channel recordings have been achieved using
homologs of BacNavs with removed voltage sensors (Shaya et al., 2011), so creating
similar Li⁺-selective mutants in these homologs would likely enable more detailed

evaluation.”

*[2] Proteins were crystalized in 100 mM NaCl and other ions. What will happen to the*
*structure, if crystalized in 100 mM Li⁺ instead of 100 mM Na⁺? Will there be any*
*change of the protein structure due to a sort of induced fit? I would like to have the*
*opinion/discussion of the authors.*

Thank you for your suggestion. We have also considered the point and already
attempted crystallization using other monovalent cations but we were unable to obtain
high-resolution crystals. Furthermore, as described in the Methods section, after
obtaining crystals under 100 mM NaCl conditions, we increased the NaCl concentration
to 2.5 M for the diffraction experiments.

This increase in NaCl concentration greatly improved the resolution. Conversely, when
we tried a similar concentration increase using other monovalent cations, such as K or
Li, the diffraction pattern completely disappeared. Based on these results, we believe
that this phenomenon is not due to resolution improvement resulting from dehydration
caused by increased salt concentration, but rather that Na ions exert a beneficial effect
on the protein crystal. The exact mechanism remains speculative, but we suspect it
involves a selective filter.

As described above, structures in the presence of other ions are highly intriguing, but
we have not yet obtained such structures. The change in filter structure due to ion
species is an interesting issue that we wish to pursue further, but unfortunately, we were
unable to include it in the current paper.

*Minor comments*

*[1] Line 110: NavAb: There is no explanation what this is.*

Thank you for pointing that out; the explanation for NavAb was indeed missing. We
have therefore added the following text:

(Page 6, Line 116)

“We used NavAb channel, a BacNav cloned from *Arcobacter butzleri*, which is the first
full-length structure of Nav at atomic resolution and a helpful model for understanding
structure-function relationships of Navs (Payandeh et al., 2011).”

*[2] Line 115: S178T NK: There is no explanation about NK. The explanation about NK*
*in lines 116-118 should be moved to line 114. Also, where does N49 locate in the*
*protein?*

Thank you for your advice. As you mentioned, moving the explanation of the N49K

mutation makes the text easier to understand. We have also added the following
sentences indicating the location of the N49K site in NavAb.

(Page 6, Line 118)

“The wild-type NavAb channel is activated even at very negative membrane potentials
and it requires a holding potential of -240 mV for recovery (Payandeh et al., 2011; Irie
et al., 2018). Such a hyperpolarized holding potential complicates the evaluation of
channel properties. The N49K mutation in the voltage-sensor domain has been shown to
shift the activation potential, allowing a holding potential of -140 mV to maintain
channel function (Gamal El-Din et al., 2013; Irie et al., 2018). The N49K site is distant
from the SF and does not affect ion selectivity (Figure 1B). To ensure stable currents,
we therefore introduced the N49K mutation into all NavAb constructs used in this
study.”

*[3] Figure 1: In Figure 2 there is a description about the solutions. Here the*
*intracellular solution is not described.*

We apologize for this omission. We have added details regarding the intracellular
solution as follows:

(Page 32, Line 895)

“Recordings of the whole-cell current responses were obtained using the ramp protocol
with the solutions of 150 mM $[\text{Li}^+]_{\text{out}}$ and 150 mM $[\text{Na}^+]_{\text{in}}$.”

*[4] Line 131: Where does T206 locate in the protein? Although it is written in line 159*
*that "T206 is located in the channel lumen", it should be described here.*

Thank you for pointing that out. We should have mentioned the location of Thr206 in
the first instance. We have modified the text accordingly as follows.

(Page 6, Line 145)

“NavAb Thr206 is located within the channel lumen, which is distant from Ser178, so
the T206A mutation was not expected to influence the effects of mutations at Ser178
(Figure 1B).”

(Page 8, Line 193)

“As mentioned earlier, the mutation at Thr206, far from Ser178, is not thought to affect
the mutation at Ser178.”

*[5] Figure 2 and related text: It is better to state here that the leak current subtraction*
*was done online, although it is described in the method section.*

Thank you for your advice. Following your suggestion, we have added a note

regarding the leak current subtraction to the legend of Figure 1 as follows:

(Page 32, Line 899)

“In this figure and subsequent electrophysiological measurements shown in in Figures 2,
3, 9, cancellation of the capacitance transients and leak subtraction were performed
using a programmed P/10 protocol delivered at -140 mV.”

*[6] Figure 3: Indication of extracellular solutions, K^+ , Cs^+ , Ca^{2+} is too small and hard*
*to read. It is better to indicate at the whole top.*

Thank you for your feedback. As you pointed out, the font size and layout should
indeed be improved for better readability. Additionally, we have corrected the erroneous
Ca^{2+} concentration of extracellular solution in this Figure. Furthermore, some font sizes
in other figures were also too small. We have therefore revised the placement of the
extracellular solution labels for Figure 3 and increased the font size of Figures 1C and
3A-D.

*[7] Figure 4: An explanation about the green particle is missing.*

We apologize for omitting annotations. The green particles indicated calcium ions. We
added the following sentence to the legend of Figure 4 (which was moved to Figure 6 in
the updated manuscript).

(Page 34, Line 961)

“The green particles indicated the calcium ions.”

*[8] Figure 4 and related text: Isn't is necessary to briefly describe here that the protein*
*was crystalized in Na^+ , not in Li^+ , although the detail is described in the method.*

As you pointed out, it is indeed important to explicitly state that this is not a Li^+
condition. We have added the following sentence to the main text and changed the title
of Figure 4 (which was moved to Figure 6 in the updated manuscript):

(Page 8, Line 195)

“To obtain high-resolution diffraction patterns, crystallization in the presence of Na^+
and diffraction experiments in high- Na^+ solutions were essential. Unfortunately, we
were unable to obtain protein crystals under Li^+ conditions without Na^+ , and when the
crystals obtained under Na^+ conditions were transferred to a Li^+ -containing solution, the
diffraction patterns disappeared in the diffraction experiments.”

The title of Figure 6

(Page 33, Line 950):

“The selectivity filter structure of NavAb N49K, S178T^{NK}, S178A^{NK}, and S178G^{NK}
mutants in high sodium conditions.”

*[9] Figure 6: "Arg185 interaction was lost in S178G NK". Interaction with what? Does*
*it mean interaction between Arg185 and Met181? I cannot find clear differences from*
*other mutants. Also, I cannot understand the relationship with the description in lines*
*206-210, "similar interactions were observed in all mutants"*

Thank you for pointing this out. The sentence in the end of figure 6 (which was moved
to Figure 8 in the updated manuscript) legend was leftover text from an older draft. We
have removed it. We apologize for any confusion this may have caused.

The ‘similar interaction’ in lines 206-210 of the original manuscript refers to the
interaction network involving Arg185, Gln172, and Glu177.

For clarity, we will rewrite “similar interaction” has been used instead as a means to
specify the interaction network involving these three residues.

*[10] Discussion: Relevant figures could be cited in each part to help readers'*
*understanding.*

Thank you for your valuable suggestion. Indeed, the initial draft of the Discussion
section may be confusing due to insufficient figure citations. We have now added
appropriate figure references throughout the section to improve clarity as follows.
(Page 12, Line 340)

“The most potent electrostatic interaction site of NavAb is comprised of the side chain
of Glu177, which is only negatively charged in the SF (Figure 1C) (Payandeh et al.,
2011).”

(Page 12, Line 346)

“In NavAb S178A^{NK/TA} and S178G^{NK/TA} (LivAb^{NK/TA}) mutants, which had greatly
improved Li⁺ selectivity compared to N49K/T206A (Figure 2E and Table 1), the loss of
the hydration water exchange site formed by the side chain of Ser178 would expose
Glu177 directly to the bulk solution (Figures 6C, D and 11).”

(Page 12, Line 355)

“In addition, in the NavAb S178G (LivAb), with the glycine mutation widening the SF
vestibule (Figures 6D and 11), ions from bulk solution would be able to easily approach
the exposed electrostatic site directly, leading to a reversal of Li⁺ and Na⁺ permeability.”

(Page 13, Line 372)

“In contrast, in the current study we found that the S178G mutation of NavAb affected
only Li⁺ selectivity, not Ca²⁺ selectivity (Figures 4A, B and Table 1).”

(Page 13, Line 374)

“However, all the NavAb M181A mutants were converted into non-selective channels
with moderate Li⁺ permeability, equivalent to Na⁺, and approximately 10-fold Ca²⁺
selectivity (Figures 9B, C, and Table 2).”

*[11] Method, line 371: There is no description about the error of the clamped voltage*
*due to series resistance, and its compensation. How about the resistance of the patch*
*pipette? How and to which extent was series resistance compensated? For example, in*
*Figure 2B right panel, the amplitude of the inward current is over 10 nA, and the error*
*is obviously very serious. Please justify that the error does not significantly affect the*
*evaluation of the reversal potentials (zero current membrane potential) which are most*
*critical in this study.*

Thank you for your comments on the electrophysiological recording conditions. As
you noted, proper control of series resistance is crucial for accurately evaluating ion
selectivity. Pipette resistance ranged from 1.9 to 5.0 MΩ, and series resistance was
compensated for in each current measurement and maintained below 10 MΩ. As you
mentioned, large current amplitudes were observed in this study. However, according to
Ohm’s law ($V = IR$, where V is voltage, I is current, and R is resistance), the reversal
potential remains unaffected when the current is zero. Therefore, we included cells
exhibiting large current amplitudes in our dataset. Following your feedback, we have
added details of these procedures to the Materials and Methods section:

Section: **Electrophysiological evaluation of ion selectivity**

(Page 7, Line 162)

“Relatively large currents up to 10 nA were generated in each cell, but series resistance
did not affect the membrane voltage at zero nA, so it was possible to precisely measure
the reversal potential.”

Section: **Electrophysiological measurement in insect cells**

(Page 16, Line 486)

“The pipette current was zeroed before forming a seal with each cell to account for the
2.25 mV liquid junction potential. Series resistance was compensated and maintained
below 10 MΩ before each current recording.”

**Reviewer #2**

*This manuscript investigates the defining characteristic of sodium channels, the*
*selective permeation of Na⁺. There are a number of structures reported for Na⁺*
*channels. These studies have clearly defined the architecture of the selectivity filter, the*
*ion binding sites in the channel. Many mutational studies on the ion binding sites have*
*also been carried out. Even with these structural and functional studies, the mechanistic*
*basis for the selective permeation of Na⁺ is not yet firmly established.*

*Here the authors focus on a Ser residue, Ser178, in the selectivity filter sequence. They*
*carry out substitutions at this Ser residue and show that the Ser178Gly substitution*
*shows a greater selectivity for Li compared to Na⁺. The mutant does not alter the ability*
*of the channel to discriminate between Na⁺ and K⁺, Cs⁺ or Ca²⁺. This indicates a clean*
*switch in selectivity of the channel from Na⁺ to Li⁺. The authors report the crystal*
*structures of the Ser179 mutants, which were determined to a moderate resolution. The*
*structures of the Ser179 mutants are basically identical to the wild type channel. These*
*structural and functional studies of the Li⁺ selective variant are the strengths of the*
*manuscript.*

*Ser178 is at the mouth of the channel pore and the authors suggest that the lack of the*
*side chain in the Ser178Gly mutant leads to an increase in water molecules that are*
*present around the mouth. The authors make this claim based on the structural data, but*
*the evidence presented is not very convincing as the water electron density observed is*
*weak. The authors need to provide more convincing evidence for why water was*
*modelled at these weak puffs of electron density seen.*

Thank you for your suggestion. As you pointed out, there is no sufficient reason to
place ‘water molecules’ here, but we believe some molecule is present in the entrance of
SF. Therefore, to avoid the misunderstanding that these are water molecules, we will
stop placing particles. Accordingly, we also change the main text and the legend of
Figure 4 (which was moved to Figure 6 in the updated manuscript).

Section: **Crystal structure of mutants**

The final paragraph was completely rewritten with the above points in mind, as
follows:

(Page 8, Line 206)

“The 178th residue is located at the entrance of the ion pathway. Met181 was located
on the bulk solution side of the 178th residue. In the NavAb N49K structure, small
electron density was observed in this vicinity in the bulk solution (Figure 6, arrow). In

other lithium-selective mutants, larger electron densities were observed in a similar
position (Figure 6, arrowhead and asterisk). These electron densities could come from
ions and water molecules contained in the solution. Due to resolution limitations, it is
difficult to determine their origin, but it is conceivable that some solution molecules
tend to remain in this vicinity. We called these electron densities ‘vicinal electron
densities’. In the S178T^{NK} and S178A^{NK} mutants, the vicinal electron densities were
very close to Met181.” “Furthermore, in the S178A^{NK} mutant, it appears to be one of
the rotamers of the Met181 side chain (Figure 6C). The vicinal electron densities in the
S178G^{NK} mutant, the highest Li⁺-selective mutant, were the largest and closest to the
entrance of the ion pathway among the other mutants (Figure 6D: asterisk). The
glycine mutation created a large space around the 178th residue position due to the
absence of side chains (Figure 6D). The high Li⁺ selectivity of the S178G mutant
suggested that this wider entrance of the selectivity filter contributes to lithium
selectivity. S178A^{NK} was the second-highest Li⁺- selective mutant. The S178G and
S178A mutations eliminated the first hydration water exchange site of the SF formed
by the hydroxyl groups of Ser178. The smaller number of hydration water-exchange
sites would be a candidate for the Li⁺ selectivity promoter. At the same time, smaller
side-chain mutations exposed the negatively charged side chain of Glu177, and this
exposed and negatively charged side chain would also be a candidate for the Li⁺
selectivity promoter.”

(Page 33, Line 958)

“Black arrows indicate the small vicinal electron densities. Black arrowheads
indicate the large vicinal electron density connected to the Met181 side chain. An
asterisk indicates the large vicinal electron density isolated from the Met181 side
chain.”

*However, a potential change in the hydration at the mouth of the pore and the idea*
*that there are fewer sites at which the water molecules hydrating the permeating Li⁺ ion*
*are exchanged for channel ligands, is an interesting idea for why the mutant channel*
*may have changed from being Na⁺ selective to Li⁺ selective. This is however a*
*hypothesis and should be clearly defined as such and is more appropriate in the*
*discussion and not in the results.*

Thank you for pointing that out. We have come to realize that the initial draft did not
clearly distinguish between the actual results and our speculation. In the revised version,
we have carefully restructured the text to ensure that these aspects are clearly and
accurately conveyed.

*The additional mutational data presented in the manuscript on M181A and Q127N do*
*not provide any additional clarity.*

As you pointed out, the results of the mutational analysis of M181A and Q127N do not
support our hypothesis. However, we believe Met181 and Gln172 play crucial roles in
selectivity and ion permeability, though our paper has not yet reached the stage of
obtaining definitive evidence. Therefore, while the discussion remains incomplete, we
wish to retain the discussion on these two residues as they are expected to become
increasingly important in future studies. Accordingly, we have revised the discussion on
these two residues to be more concise and to make the results easier to understand.

*The major strength of the manuscript is the structure of a channel with selectivity*
*switched from Na⁺ to Li⁺. The structure data presented will be useful for computational*
*studies on the mechanism of ionic selectivity in Na⁺ channels.*

*Minor points:*

*The authors claim that Li⁺ has a stronger positive charge than Na⁺ (Line:306). It should*
*be changed to say that Li⁺ has a higher charge density than Na⁺.*

Thank you for pointing that out. You are absolutely correct, and we have accordingly
revised the terminology.

*An Fo-Fc omit map with the residue omitted should be used to show the fit of the side*
*chain to the electron density.*

Following your suggestion, we calculated an omit map excluding amino acid residues
surrounding the selective filter (residues 172 to 185).

This clearly demonstrated the difference in electron density of the side chain at residue
178 introduced by the mutation, so we added it as a Fig 5 in the updated manuscript.

Thanks to your feedback, we were able to enhance the clarity of our paper.

**Reviewer #3**

*In this work, Madea et al. report that mutations of S178 in NavAb enhanced*
*Li-selectivity in resulting channels, with S178G converting NavAb into a Li-selective*
*channel. The authors also solved structures of NavAb S178 mutants, suggesting that*
*S178 mutations remove the first hydration water exchange site in the selectivity filter*
*(SF) of NavAb to enhance Li-selectivity by three mechanisms: reducing the number of*
*hydration water exchanges during Li⁺ permeation, allowing stronger electrostatic*
*interaction between hydrated Li⁺ and Glu177 in the SF, and altering the supportive*
*network behind the SF. The Li-selective NavAb constructs developed in this work could*
*serve as an important tool to study Li⁺-selectivity in Nav channels. However, the*
*differences between the structures of WT and S178-mutated channels are quite subtle,*
*which makes some conclusions in the paper less well-founded. Below are my specific*
*comments.*

*1. The reversal potential was determined by a voltage ramp protocol in this study. It*
*may be necessary to verify the results with instantaneous I-V recordings, particularly in*
*cases where the reversal potential lies within the deactivation voltage range.*
*Alternatively, the reversal potential could be measured with Na on the extracellular side*
*and less-permeable cations on the intracellular side, ensuring the reversal potential is*
*within the activation voltage range of NavAb.*

Thank you for your thorough review of this study and for your insightful comments.
As you correctly pointed out, the evaluation potentials overlapped with the deactivation
membrane voltages in the measurements of K⁺- and Cs⁺-selectivity, making it difficult
to accurately determine the reversal potentials. However, the key point we emphasize in
this paper is that the Ser178 mutation in NavAb markedly enhances Li⁺ selectivity alone,
while the selectivity for other cations remains extremely low. We believe that the
present results well support this conclusion. If we pursue a more detailed investigation
of the changes in K⁺-and Cs⁺-selectivity in the future, we will consider adopting the
method you suggested—using Na⁺ as the external solution and K⁺ or Cs⁺ as the internal
solution. We sincerely appreciate your valuable advice.

We also recognize that it is essential to mention the overlap between the evaluation
potentials and the deactivation membrane voltages in the main text. Accordingly, we
have added the following sentence to the *Results* section:

**Section: Electrophysiological evaluation of ion selectivity**

(Page 7, Line 174)

“Due to the deactivation induced by the negative membrane potential, it was difficult to
perform precise measurements of the reversal potential in these ionic environments.
However, the reversal potential around the deactivation membrane potential was
sufficient to show that the permeability to these ions was extremely low in all
constructs.”

*2. How were the series resistance error and liquid junction potential compensated in the*
*experiment?*

Thank you for your comments on the electrophysiological recording conditions. As you
pointed out, proper control of series resistance and liquid junction potential is needed
for reversal potential measurements. Series resistance was compensated before each
current measurement. Liquid junction potential was compensated when the pipette
internal solution came into contact with the external bath solution. In response to your
comment, we have added details of these procedures to the *Materials and Methods*
section:

**Section: Electrophysiological measurement in insect cells**

(Page 16, Line 486)

“The pipette current was zeroed before forming a seal with each cell to account for the
2.25 mV liquid junction potential. Series resistance was compensated and maintained
below 10 MΩ before each current recording.”

*3. It was mentioned in the Methods that "Cells with a smaller leak current than 1nA*
*were used for data collection" (page 21, line 401). Considering the amplitude of sample*
*currents shown in the figures, a threshold of 1nA for leak current seems excessively*
*large.*

Thank you for pointing this out. The phrase ‘1 nA or less’ in the original manuscript
referred to the cutoff value used during measurements. The actual leak currents for all
cells used in the dataset are summarized as follows, showing both the average and
maximum values. In this case, the error introduced by leak current into the reversal
potentials is negligible and does not affect data interpretation. Therefore, we believe that
the measurements were performed appropriately. We have rewritten the *Materials and*
*Methods* section accordingly.

Section: **Electrophysiological measurement in insect cells**

The final paragraph was rewritten with the above points in mind, as follows:

(Page 16, Line 488)

“Cells with leak current < 800 pA were used for data collection, and the mean leak
current ranged from 110 to 190 pA for each construct.”

*4. One major conclusion from this work is that Li⁺ selectivity is enhanced by removing*
*the first water exchange site and widening of extracellular entrance of SF in NavAb.*
*However, the results of S178T (increased Li⁺ selectivity, seemingly unchanged water*
*exchange sites, and narrower extracellular entrance) appear to contradict other*
*conclusions. The authors should include a more detailed discussion about this.*

Thank you for your comment. As you mentioned, the increase of Li⁺ selectivity in the
S178T mutant should not be overlooked. However, we believe the key finding of this
study is that the NavAb Ser178 mutation created the channels with Li⁺ selectivity
exceeding Na⁺ selectivity. In the S178A^{NK/TA} and S178G^{NK/TA} mutants, where P_{Li}/P_{Na}
exceeds 1, the monovalent cation selectivity shifted from the Eisenmann sequence Seq.
X (Na⁺ > Li⁺) to Seq. XI (Li⁺ > Na⁺). The order of monovalent cation selectivity in the
Eisenman sequence is determined by electrostatic interactions. This leads to the
conclusion that changes in the electrostatic interactions between the exposed Glu177
and ions are most crucial for Li⁺ selectivity.

Furthermore, in response to your suggestion, we performed Dunnett’s test to compare
Li⁺ selectivity among N49K/T206A (which contains Ser178) and its S178T, S178A, and
S178G mutants, to analyze S178T in more detail. The statistical results showed that the
alanine and glycine mutants differed significantly from N49K/T206A, whereas S178T
did not show a significant difference. This supports our hypothesis that the alanine and
glycine substitutions enhance Li⁺ selectivity through reduced hydration water exchanges
and an enlarged SF entrance. However, S178T also exhibited an upward trend in Li⁺
selectivity, suggesting that further investigation is warranted despite the lack of
statistical significance. Met181 is a likely candidate contributing to the increased Li⁺
selectivity of S178T; accordingly, we have added a description of the involvement of
Met181 in Li⁺ selectivity to the Results section.

Section: **Electrophysiological evaluation of ion selectivity**

In response to your sharp comment, we have added the following text:

(Page 7, Line 168)

“A slight positive shift was also shown by S178T^{NK/TA}, but there was no significant
difference of P_{Li}/P_{Na} compared to N49K/T206A (Figure 2E).”

(Page 8, Line 214)

“In the S178TNK and S178ANK mutants, the vicinal electron densities were very close
to Met181.”

*5. Could the main chain oxygens of S178G form a water exchange site in the mutant?*

Thank you for your question. In the crystal structure, the main chain oxygens of
S178G face the wide vestibular space of the SF, but are positioned away from the ion
conduction pathway. Although it may potentially interact with ions, it is therefore
unlikely to do so during ion permeation through the SF.

*6. In Fig. 6, it is difficult to discern how the network involving M181 in the S178G*
*structure differs from that in other NavAb constructs. Even if the difference in M181*
*observed in the S178G structure is valid, the functional result that M181A only changes*
*the Ca²⁺-selectivity of NavAb does not support authors' statement that "The difference*
*in Met181 observed in S178G may, therefore, be responsible for improving Li*
*selectivity" (page 12 line 211) .*

Thank you for your comment. First, your comment made me realize that Figure 6 of
old manuscript had visibility issues. Therefore, we omitted the electron density in this
figure (which was moved to Figure 8 in the updated manuscript). Considering the
surrounding context, I believe “Gln172” is more appropriate than ‘Met181’ in the
sentence “The difference in Met181 observed in S178G may, therefore, be responsible
for improving Li selectivity”. So, we have revised the section as follows:

(Page 9, Line 252)

“This interaction network involving the three residues was also observed in all mutants
(Figures 8 A-D). Although resolution was limited, the hydrogen bond between the main
chain of glutamate 177 and the side chain of glutamine 172 appeared closer in the
S178G mutant than in other variants (Figure 8D). We focused on glutamine 172, which
mediates the hydrogen-bond network, as a regulator of ion permeability and selectivity.
Gln172 of NavAb N49K/T206A and S178GNK/TA were mutated to evaluate their role
in regulating Li⁺ selectivity. The asparagine side chain is shorter than that of glutamine
by one methyl group deletion, so the Q172N mutation was expected to weaken or lose
the hydrogen bond networks with Glu177 and Arg185. Neither the NavAb Q172N
mutant conducted any current. This fatal loss of function due to a slight difference in
side-chain length suggested a critical role for Gln172 of NavAb in ion permeability and
stability of the SF.”

Furthermore, the changes occurring around Met181 were difficult to discern, so we have

added pictures of the ion pore viewed from the extracellular side to Figure 4 (which was
moved to Figure 6 in the updated manuscript).

This should clearly demonstrate the increased electron density near Met181 in the
Li-selective mutant.

To highlight the importance of Met181, we added the following description:
(Page 8, Line 214)

“In the S178TNK and S178ANK mutants, the vicinal electron densities were very close
to Met181.”“Furthermore, in the S178ANK mutant, it appears to be one of the rotamers
of the Met181 side chain (Figure 6C).”

(Page 13, Line 380)

“The Met181 of NavAb is Ser in NaChBac, which is one of the representative BacNavs
(Ren et al., 2001; Yue et al., 2002). Ser178 is highly conserved in BacNavs, but a
similar glycine mutation of other BacNavs, such as S178G of NavAb, may not improve
Li⁺ selectivity.”

We believe Met181 and Gln172 play crucial roles in selectivity and ion permeability,
though our paper has not yet reached the stage of obtaining definitive evidence.

Therefore, while the discussion remains incomplete, we wish to retain the discussion on
these two residues as they are expected to become increasingly important in future
studies. Accordingly, we have revised the discussion on these two residues to be concise
and to make the results easier to understand.

*7. The differences among the structures of S178 mutants are quite subtle. This is not*
*unexpected given that the structures of these NavAb constructs with similar selectivity*
*for Na were determined in NaCl. Is it possible to solve the structures in LiCl?*

Thank you for your suggestion. We have also considered the point and attempted
crystallization under LiCl condition. However, we were unable to obtain high-resolution
crystals with lithium ion. Furthermore, as described in the methods section, after
obtaining crystals under 100 mM NaCl conditions, we increased the NaCl concentration
to 2.5 M for the diffraction experiments.

This increase in NaCl concentration significantly improved the resolution. Conversely,
when we tried a similar concentration increase using other monovalent cations, such as
K⁺ or Li⁺, the diffraction pattern completely disappeared. Based on these results, we
believe that this phenomenon is not due to resolution improvement resulting from
dehydration caused by increased salt concentration, but rather that Na ions exert a
beneficial effect on the protein crystal. The exact mechanism remains speculative, but

we suspect it involves a selective filter.

As described above, structures in the presence of other ions are highly intriguing, but
we have not yet obtained such structures.

The change in filter structure due to ion species is an interesting issue that we wish to
pursue further, but unfortunately, we were unable to include it in this paper.

*8. The quality of the English writing in this paper could be improved, and seeking*
*professional editorial assistance might be beneficial.*

Thank you for the comment. We have worked closely with a professional English editor
in the revision of our paper. Subsequently, there are a number of miscellaneous changes
to the text throughout the manuscript in addition to the comment-specific changes.

January 16, 2026

Dr. Katsumasa Irie
Wakayama Medical University
25-1, Shichibancho
Wakayama, Wakayama 640-8156
Japan

Re: 202513855R1

Dear Dr. Irie,

Thank you for submitting your manuscript, entitled "Structure-function analysis of the lithium-ion selectivity of the voltage-gated sodium channel" to JGP. Your manuscript has now been seen by 3 reviewers, whose comments are appended below. You will see that the reviewers are happy with the changes you have made. I am attaching a file that lists a few places in the manuscript that might benefit from some additional clarification.

We appreciate the new statistical information that you have provided, but Supplementary Table 1 provided some discussion among the Associate Editors. We understand the use of Dunnett's Test, but found the values for d and t and how that led to your particular N.S. or $P < 0.01$ a bit confusing. An outside statistical expert was consulted, who pointed out that the absolute value of the test statistic t is the value compared to the critical values, d .

It seems likely that Supplementary Table 1 will not necessarily be helpful to most readers. However, it is our understanding that most statistics packages would allow Dunnett's test to return an explicit p -value. Therefore, our recommendation is that you simplify Supplementary Table 1 and simply refer the specific p -values, as is JGP policy.

Again, thank you for responsive to the reviewers and for submitting your work to JGP.

I will look forward to submission of a revised manuscript that addresses the minor points on the attached file and the statistics question. It will most likely not be sent out for review again. In addition, please do not hesitate to contact me (via the editorial office) if you feel that a discussion of the reviewers' and editors' comments would be helpful.

Please submit your revised manuscript via the link below, along with a point-by-point letter that details your response to the reviewers' and editor's summary, as well as a copy of the text with alterations highlighted (boldfaced or underlined). If the article is eventually accepted, it would include a 'revised date' as well as submitted and accepted dates. If we do not receive the revised manuscript within one year, we will regard the article as having been withdrawn. We would be willing to receive a revision of the manuscript at a later time, but the manuscript will then be treated as a new submission, with a new manuscript number.

Please pay particular attention to recent changes to our instructions to authors in the following sections: Data presentation, Blinding and randomization and Statistical analysis, under Materials and Methods, as shown here: <https://rupress.org/jgp/pages/submission-guidelines#prepare>. Re-review will be contingent on inclusion of the required information (including for data added during revision) and demonstration of the experimental reproducibility of the results. Also, To improve the reproducibility of published content, we have partnered with SciScore. Authors are prompted in eJP to copy and paste the Materials and Methods section of their manuscript for a SciScore assessment when submitting their revised manuscript. Authors are encouraged (not required) to further revise their Materials and Methods if the SciScore is below 4. More information can be found here: <https://rupress.org/jgp/pages/submission-guidelines#sciscore>.

Please note, JGP now requires authors to submit Source Data used to generate figures containing gels and Western blots with all revised manuscripts (when applicable). This Source Data consists of fully uncropped and unprocessed images for each gel/blot displayed in the main and supplemental figures. If your paper includes cropped gel and/or blot images, please be sure to provide one Source Data file for each figure that contains gels and/or blots along with your revised manuscript files. File names for Source Data figures should be alphanumeric without any spaces or special characters (i.e., SourceDataF#, where F# refers to the associated main figure number or SourceDataFS# for those associated with Supplementary figures). The lanes of the gels/blots should be labeled as they are in the associated figure, the place where cropping was applied should be marked (with a box), and molecular weight/size standards should be labeled wherever possible.

Source Data files will be made available to reviewers during evaluation of revised manuscripts and, if your paper is eventually published in JGP, the files will be directly linked to specific figures in the published article.

Source Data Figures should be provided as individual PDF files (one file per figure). Authors should endeavor to retain a minimum resolution of 300 dpi or pixels per inch. Please review our instructions for export from Photoshop, Illustrator, and PowerPoint here: <https://rupress.org/jgp/pages/submission-guidelines#revised>

Whilst you are revising your manuscript, we ask that you consider whether you have any artwork that might be suitable for the cover of JGP. Microscopy images are particularly good for cover artwork, but other types of image can be very effective, so we encourage you to be creative. Please don't restrict yourself to images from the paper; an image that is relevant to the work described would be just as suitable. Images should be a minimum resolution of 300 dpi. To see recent examples, visit the following page and click on 'Show covers? Yes': <https://jgp.rupress.org/content/by/year>

Thank you for submitting your interesting research to JGP.

Please submit your revised manuscript, and any associated files, via this link:

Link Not Available

Sincerely,

Christopher Lingle, Ph.D.

On behalf of Journal of General Physiology

Journal of General Physiology's mission is to publish mechanistic and quantitative molecular and cellular physiology of the highest quality; to provide a best-in-class author experience; and to nurture future generations of independent researchers.

Reviewer #1 (Comments to the Authors):

I noticed that there were obvious mistakes in my previous comments. I consistently described Ser178 (wt) as Thr178. I am sorry for the confusion caused by my mistake, and thank the authors for correctly understanding my comments in spite of the errors.

I checked the authors' responses and the revised manuscript. The authors fully responded to the two major points of my comments. Although there are points remained to be solved in future, e.g. the structure with Li⁺, I am satisfied with the authors' responses/ explanations/ and revisions of the text. As to my 11 minor comments, the revisions are also appropriate and satisfactory. I have no other specific comments.

Reviewer #2 (Comments to the Authors):

The authors took all my concerns into account and addressed them to my satisfaction.

Reviewer #3 (Comments to the Authors):

The revised manuscript is much improved, and my previous concerns have been well-addressed. I have no further comments.

Thank you for reading the revisions to our manuscript, and we sincerely appreciate your valuable and insightful comments. In the updated manuscript, we made revisions corresponding to your comments and corrected typos. The textural changes in the updated manuscript are shown in the underlined text.

In response to your suggestions regarding the statistical analysis, we have reconsidered the methodology and presentation of Dunnett's Test. In the original manuscript, Dunnett's Test was performed using Microsoft Excel according to the statistics textbook we consulted, which did not allow for the calculation of exact p -values. Significant differences were therefore determined by comparing the test statistic (t) with the critical value (d) (Supplementary table 1 of the previous manuscript). In this revision, we have recalculated the data using Python and the SciPy package to obtain precise p -values. These p -values have been incorporated into Main Tables 1 and 2 of the updated manuscript. Consequently, we have removed Supplementary Table 1 of the previous manuscript.

Following the re-analysis using Python, the p -values for P_{Li}/P_{Na} in the N49K/T206A and S178A^{NK/TA} groups were calculated as $p=0.01$. Accordingly, the significance level in Figure 2E was updated from $p<0.01(**)$ to $p<0.05(*)$. This is likely due to a slight changes in statistical values, as the current Python-based analysis utilizes unbiased variance (with Bessel's correction), whereas our previous calculation relied on biased sample variance.

In the original manuscript, statistical values for the groups including P_X/P_{Na} was <0.05 (X represents Li, K, Cs, or Ca) were calculated using the onset potential of outward currents (i.g., K^+ and Cs^+ currents in Figure 3A). However, we recognized that comparing groups with quantifiable data against those lacking definitive values is not statistically sound. These comparisons were therefore deleted (Figures 2E, 4B, 9C, 10 and Table 1, 2 in the updated manuscript). Furthermore, as the results of Student's t -tests are already presented in Figure 10, we have removed Supplementary Table 2 to avoid redundancy.

Responses to the comments received in the PDF is detailed on the following page. We believe the manuscript has been greatly strengthened thanks to your valuable feedback. Thank you for the opportunity to improve our work. We hope the revised version meets your expectations and would appreciate your further review.

Responses to suggested minor edits

Detailed responses to each comment and the associated modifications are provided below. All the line numbers indicated in these responses are referred to in the updated version.

Line 71. It may be helpful for readers if some of the ionic differences between Na⁺ and Li⁺ are listed at this point.

In response to the comments, we have added the following text to the *Introduction* section:

(Page 3, Line 69)

“Na⁺ and Li⁺ are monovalent and single-atom cations, and the ionic radius of Li⁺ (0.60 Å) is smaller than that of Na⁺ (0.95 Å). Due to its small radius, Li⁺ has a higher positive charge density providing a strong electrostatic force than Na⁺ (Hille, 2001).”

Additionally, we revised similar explanatory text in the *discussion* section:

(Page 11, Line 322)

“Li⁺ (0.60 Å) has a smaller ionic radius than Na⁺ (0.95 Å), resulting in a higher positive charge density and a stronger electrostatic interaction with the negative dipole of water molecules and proteins. Consequently, the rate of hydration water exchange for Li⁺ is slower than that for Na⁺ (Hille, 2001).”

Line 241. “extensive residues forming ion pores” seems very unclear. Does this mean residues lining the pore? Residues behind the selectivity filter that may help stabilize the filter structure? How was it decided which residues “form ion pores”. Do you mean something like the following? “We focused on residues that either seem likely to mediate interactions with ions as they transit the pore or that are thought likely to help stabilize SF structure”

We realized the original sentence was quite ambiguous as your comment. We therefore inserted your suggestion into the manuscript.

(Page 9, Line 240)

“Next, we focused on residues that either seem likely to mediate interactions with ions as they transit the pore or that are thought likely to help stabilize SF structure (Figure 8).”

Line 261. “Neither” means there are two things being considered. “neither mutant1 nor mutant2”. Perhaps something like....

“so the Q172N mutation might be expected to weaken or abolish the hydrogen bond networks with glu177 and Arg185. In xx cells from xx transfections, we failed to observed any current with the NavAb Q172N mutant. “

Thank you for your proposal. We have incorporated your suggestion into the manuscript.

(Page 9, Line 258)

“The asparagine side chain is shorter than that of glutamine by one methyl group deletion, so the Q172N mutation might be expected to weaken or abolish the hydrogen bond networks with Glu177

and Arg185. In five cells per mutant from a single transfection, we failed to observe any current with the NavAb Q172N mutants.”

We incorporated your following suggestions into the manuscript as is. Thank you for offering more suitable sentences.

Line 265 “Effects of mutation of a pore helix residue on monovalent cation selectivity” (reflected in **Page 10, Line 266**)

Line 274. “showed little current and sometimes a current too weak (<1 nA) to evaluate reversal potential” (reflected in **Page 10, Line 274**)

Line 277. Although it may be standard in spoken communications to simply say P_{Li}/P_{Na} , that is really an abbreviation for “ P_{Li}/P_{Na} ratio” In this sentence, we would recommend “The resulting P_{Li}/P_{Na} ratios were” There are other places in the text where it would be best to also make such a change.

Thank you for pointing this out. We corrected all the corresponding words (P_{Li}/P_{Na} ratio, P_{Ca}/P_{Na} ratio, P_M/P_{Na} ratio and P_X/P_{Na} ratio) in the main text and legends.

We incorporated your following three proposals into the manuscript as is. Thank you for suggesting more appropriate sentences.

Line 326. “there will be a slower rate” (reflected in **Page 11, Line 326**)

Line 335: suggestion: “hydration water exchanges in the SF alone would not be expected to increase Li permeability” (reflected in **Page 12, Line 335**)

Line 366-367. The phrase, “Because Ca^{2+} has a strong electrostatic force than monovalent cations. “ is unclear. Should it read, “despite Ca^{2+} having a stronger electrostatic force than monovalent cations”. (reflected in **Page 12, Line 366**)

Line 426: “ P_{Li}/P_{Na} ratios” (reflected in **Page 14, Line 425**)

In the R1 submission, the image of Figure 6 was assembled before than of Figure 5. Please correct. We apologize for not noticing the mistake last time. It is corrected in this submission.

In addition to the above, the following typos were corrected.

(Page 16, the section of Electrophysiological measurement in insect cells in Materials and methods)

The sentence “Cells with a smaller leak current than 1 nA were used for data collection.” was deleted because it was replaced with the detailed measurement conditions for the electrophysiological measurement.

(Page 30)

In Table 1, P_K/P_{Na} ratio of S178A^{NK/TA} was corrected from “ 0.05 ± 0.00 ” to “ <0.05 ”

Dr. Katsumasa Irie
Wakayama Medical University
25-1, Shichibancho
Wakayama, Wakayama 640-8156
Japan

Re: 202513855R2

Dear Dr. Irie,

I am pleased to let you know that your manuscript, titled "Structure-function analysis of the lithium-ion selectivity of the voltage-gated sodium channel" is scientifically acceptable for publication in Journal of General Physiology. Formal acceptance will follow when it is modified in accordance with any editorial policy issues that may be mentioned below.

Please note items that need attention are listed at the bottom of this email (under 'manuscript formatting checklist'). Please also be sure to include a letter addressing the reviewers' comments point-by-point (if applicable) and a copy of the text with alterations highlighted (boldfaced or underlined). Your manuscript should be a double-spaced MS Word file and include editable tables, if appropriate.

Lastly, JGP requires a data availability statement for all research article submissions. These statements will be published in the article directly above the Acknowledgments. The statement should address all data underlying the research presented in the manuscript. Please visit the JGP instructions for authors for guidelines and examples of statements at <https://rupress.org/jgp/pages/editorial-policies#data-availability-statement>.

Please submit your final files via this link:
Link Not Available

Thank you for choosing to publish your research in JGP and please feel free to contact me with any questions.

Sincerely,

Christopher Lingle, Ph.D.
On behalf of Journal of General Physiology

Journal of General Physiology's mission is to publish mechanistic and quantitative molecular and cellular physiology of the highest quality; to provide a best in class author experience; and to nurture future generations of independent researchers.

Manuscript formatting checklist:

- MS Word document of text needed (including editable tables)
- MS Word document of supplemental text needed, if applicable (including figure legends and editable tables)
- Brief Statement describing supplementary information needed, if applicable (in subsection at end of Materials & Methods)
- Please include a data availability statement preceding the Acknowledgments section. Please see <https://rupress.org/jgp/pages/editorial-policies#data-availability-statement>
- Figures created at sufficient resolution and in acceptable format (including supplemental if applicable). If working in Illustrator, we prefer .ai or .eps file format. If working in Photoshop please use 600dpi/1000dpi .tiff or .psd file format. Minimum resolution at estimated print size: Minimum resolution for all figures is 600 dpi. For figures that contain both photographs and line art or text, 600 dpi is highly recommended. Figures containing only black and white elements (line art, no color, and no gray) should be 1,000 dpi. Maximum figure size is 7 in wide x 9 in high (17.5 x 22.8 cm) at the correct resolution. <https://jgp.rupress.org/fig-vid-guidelines>
- Supplemental figures, if any, conforming to same guidelines as manuscript figures (noted above)
- If images resemble one from a prior publications, the author must seek permissions (to reproduce or adapt) from the original publisher. [You can resubmit your paper while waiting to hear back from the original publisher but please keep us updated]
- All authors must complete a disclosure form prior to acceptance. A link to complete the form has been sent to all coauthors. Please provide the editorial office with updated email addresses if necessary